# Prioritizing Perception-Guided Self-Supervision: A New Paradigm for Causal Modeling in End-to-End Autonomous Driving

**Yi Huang**[12][*], **Zhan Qu**[2][*], **Lihui Jiang**[2][†], **Bingbing Liu**[2], **Hongbo Zhang**[2]

[1]The Chinese University of Hong Kong, Shenzhen
[2]Huawei Noah's Ark Lab
yihuang11@link.cuhk.edu.cn
{quzhan, jianglihui1, liu.bingbing, zhanghongbo888}@huawei.com

## Abstract

End-to-end autonomous driving systems, predominantly trained through imitation learning, have demonstrated considerable effectiveness in leveraging large-scale expert driving data. Despite their success in open-loop evaluations, these systems often exhibit significant performance degradation in closed-loop scenarios due to causal confusion. This confusion is fundamentally exacerbated by the overreliance of the imitation learning paradigm on expert trajectories, which often contain unattributable noise and interfere with the modeling of causal relationships between environmental contexts and appropriate driving actions. To address this fundamental limitation, we propose Perception-Guided Self-Supervision (PGS)—a simple yet effective training paradigm that leverages perception outputs as the primary supervisory signals, explicitly modeling causal relationships in decision-making. The proposed framework aligns both the inputs and outputs of the decision-making module with perception results—such as lane centerlines and the predicted motions of surrounding agents—by introducing positive and negative self-supervision for the ego trajectory. This alignment is specifically designed to mitigate causal confusion arising from the inherent noise in expert trajectories. Equipped with perception-driven supervision, our method—built on a standard end-to-end architecture—achieves a Driving Score of 78.08 and a mean success rate of 48.64% on the challenging closed-loop Bench2Drive benchmark, significantly outperforming existing state-of-the-art methods, including those employing more complex network architectures and inference pipelines. These results underscore the effectiveness and robustness of the proposed PGS framework, and point to a promising direction for addressing causal confusion and enhancing real-world generalization in autonomous driving.

## 1 Introduction

Autonomous driving, as a significant application of AI, has made impressive advancements in recent years. End-to-end neural networks, which allow vehicles to make decisions directly from raw sensor signals, are considered capable of overcoming the cumulative error issues inherent in traditional modular approaches and offer the potential to scale with vast amounts of data. In mainstream systems, perception, prediction, and planning tasks are integrated into a single network. The planning module uses explicit or implicit representations of the environment provided by perception to plan the future

---

[*]Equal contribution.
[†]Corresponding author.

39th Conference on Neural Information Processing Systems (NeurIPS 2025).

behavior, with human or expert trajectories being used as the target of training. Researchers have made significant efforts to leverage large-scale human driving data to enable models to learn the relationship between environmental context and vehicle behavior.

As a classic paradigm for end-to-end systems, imitation learning became prominent alongside early benchmarks such as nuPlan [3], Argoverse [4], Oxford RobotCar [2]. These benchmarks typically provide open-loop metrics, with L2 error between predicted and ground-truth trajectories as the key indicator. Consequently, researchers focus on designing complex network architectures, incorporating multi-modal sensor information, and using imitation learning objective functions aligned with these metrics, to enhance the model's ability to fit expert trajectories. However, recent studies have shown that trajectory fitting in open-loop evaluation cannot accurately reflect system performance in real-world scenarios [15, 19, 28]. In closed-loop simulation tests, pure imitation learning models often show significant degradation in safety, comfort, and feasibility in complex scenarios. This inability to generalize in real-world environments has become a major challenge for end-to-end systems.

Among the factors affecting the closed-loop performance, the most significant is causal confusion. Causal confusion refers to the model's inability to associate driving behavior with the primary causal factors in the environment, instead linking it to other noise factors. Although recent end-to-end approaches have reduced input noise by using sparse instance-level representation [14] of the environment, these methods still fail to fully address this problem. In this paper, We identify causal confusion as an unavoidable byproduct of the imitation learning framework, stemming from its reliance on suboptimal expert data. Expert or human trajectories often contain noise from factors like driving style, time of day, or control errors, making them suboptimal supervision targets. Learning from such noisy signals weakens the model's ability to capture true causal relationships. We argue that causal confusion stems not just from imperfect inputs, but more critically from noise in the supervision itself.

Unlike prior approaches that treat perception and prediction modules merely as feature extractors, we propose a framework leverages their outputs as primary supervision signals for decision-making. By aligning both the inputs and outputs of the decision-making module with perception outputs, our perception-guided self-supervision paradigm exhibits stronger causal modeling capabilities in closed-loop evaluations than pure imitation learning. Specifically, we introduce three novel self-supervision mechanisms: Multi-Modal Trajectory Planning Self-Supervision (MTPS), Spatial Trajectory Planning Self-Supervision (STPS), and Negative Trajectory Planning Self-Supervision (NTPS). MTPS and STPS utilize lane centerlines to enforce topological constraints and support multimodal decision-making across available lanes. NTPS incorporates the predicted future trajectories of dynamic agents as negative supervision to guide the ego vehicle away from potential collisions. In this framework, human expert trajectories are used to filter or regularize self-supervision targets when perception-based guidance is unavailable.

In summary, we propose an innovative training paradigm for end-to-end autonomous driving systems, which does not rely on specialized network designs but emphasizes the use of perception-guided self-supervision as the main learning objective. Our contributions include the following:

1. **Multi-Modal Trajectory Planning Self-Supervision as Target Lane Selection:** We reformulate multi-modal ego decision-making as a target lane selection problem based on lane perception. This approach enhances the system's ability to associate surrounding obstacles and available lanes with appropriate driving decisions, thereby improving the performance of lane-change planning.

2. **Spatial Trajecotry Planning Self-Supervision based on lane centerline:** We take the lane centerline outputted from perception module as a spatial trajectory without temporal dependency, and use them as the primary learning target for planning ego trajectory. This design effectively reduces lane departures and mitigating causal confusion induced by inconsistent and noisy expert demonstrations.

3. **Negative Trajectory Planning Self-Supervision from Dynamic Objects' Future bounding box:** Our framework selects and utilizes the predicted future trajectories of surrounding agents as negative supervision signals for ego trajectory learning, enforcing non-overlapping constraints between future bounding boxes. This facilitates the learning of interactions with dynamic agents and reduces collision risk.

4. We made minimal modifications to a simple end-to-end network architecture to adapt and validate our proposed self-supervision training paradigm. In experiments on the challenging closed-loop benchmark, Bench2Drive, the self-supervised model outperformed the pure imitation learning version of the same architecture and recent works using more complex network structures and pipelines by a large margin.

## 2 Related Work

End-to-end autonomous driving aims to generate planning trajectories directly from raw sensors. In the field, advancements have been categorized based on their evaluation methods: open-loop and closed-loop systems. We reviewed representative works based on these two evaluation schemes in the first and second subsections, and summarized existing techniques and improvements addressing the causal confusion in the third subsection.

### 2.1 Open-Loop End-to-End Driving Methods

In open-loop systems, UniAD [8] proposes a unified framework that integrates full-stack driving tasks with query-unified interfaces, enhancing task interaction. VAD [14] improves planning safety and efficiency, as demonstrated by its performance on the nuScenes dataset. SparseDrive [23] uses sparse representations to mitigate information loss and error propagation in modular systems. ParaDrive [26] organizes perception, motion prediction, and planning tasks in a parallelized architecture during training, retaining only the planning module in inference. This approach improves planning performance and significantly reduces runtime latency

### 2.2 Closed-Loop End-to-End Driving Methods

Existing works (e.g., BEVPlanner [19]) have found that metrics like L2 error and collision rate used in open-loop evaluations do not comprehensively reflect model performance in real-world scenarios. As a result, more approaches are being proposed for closed-loop evaluation. VADv2 [5] advances vectorized autonomous driving by generating action distributions for vehicle control, achieving outstanding performance on the CARLA Town05 benchmark. Transfuser [6] uses transformer modules at multiple resolutions to fuse perspective and bird's-eye view feature maps, outperforming prior work on the CARLA leaderboard. Hydra-MDP [17] employs knowledge distillation from both human and rule-based teachers to train the student model, enabling the selection of the trajectory with optimal overall performance and securing first place in the Navsim challenge. DriveTransformer [13] delves into task parallelism and sparse representation in architecture design, significantly improving driving scores and success rates on the Bench2Drive benchmark.

### 2.3 Techniques Addressing Causal Confusion in Autonomous Driving

Causal confusion has been a persistent challenge in imitation learning. In end-to-end driving, ChauffeurNet [1] addresses this issue by using past ego-motion as intermediate BEV abstractions, and randomly dropping them during training. PrimeNet [25] improves performance by incorporating predictions from a single-frame model as additional input to a multi-frame model. DriveAdapter [11] mitigates the influence of noisy perception outputs by training a strong planner with privileged perception information, and aligning perception model output with the planner's input through an adapter. RAD [7] proposes a 3DGS-based closed-loop reinforcement learning framework, which uses specialized rewards to guide the policy in understanding real-world causal relationships more effectively. These approaches primarily aim to mitigate causal confusion by suppressing noise in the inputs to the planning module. In this paper, we propose a novel perspective and an innovative training paradigm, where perception-guided self-supervision plays a key role in addressing causal confusion by aligning the input and output of the planning module.

## 3 Method

In this section, we introduce a Prioritizing Perception-Guided Self-Supervision training paradigm built upon a typical end to end architecture. On one hand, sparse instance-level features extracted from perception are used as inputs to the unified decision and prediction module, helping minimize

input noise. On the other hand, the perception output is directly employed for self-supervision of the planning process. This alignment plays an important role in helping the planner learn causal relationships.

## 3.1 End-to-End Network Baseline

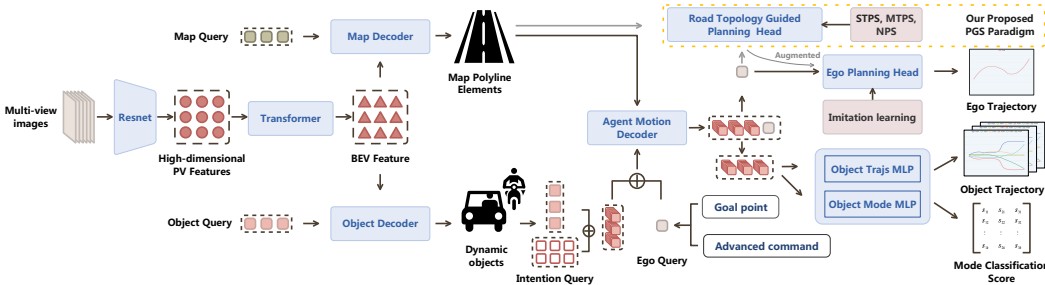

Figure 1: Overall Model Architecture. (1) Perception network provides map elements and future motions of dynamic agents. (2) The dashed box presents the proposed PGS, which generates three self-supervised signals from perception outputs to enhance causal reasoning in ego planning.

Our baseline architecture is simple and similar to VAD [14]. As illustrated in Figure 1, multi-view images are encoded into high-dimensional features in perspective view using a Resnet-based encoder. BEVFormer [18] then transform the features into BEV (Bird's Eye View) space as $F_{\text{BEV}}$.

For task-specific applications, learnable embeddings are used to query the BEV features, extracting sparse representations of the environment. The map query interacts with the BEV features to decode static road topology such as lane markings, lane centerlines, sidewalks, and other structures. The object query decodes information about dynamic objects (pedestrians, cyclists, and vehicles), including their positions, sizes, orientations, and velocities at the current timestep. The perception process can be formally expressed as follows:

$$q_i' = \text{PerceptionDecoder}\big(q_i, kv = F_{\text{bev}}\big), \quad q_i \in Q_{\text{map}} \cup Q_{\text{obj}}, \tag{1}$$

$$\hat{y}_i^{\text{map}} = \text{MLP}_{\text{map}}\big(q_i'^{\text{map}}\big) = \big(\{p_j^{(i)}\}_{j=1}^{N_{\text{points}}}, c^{(i)}\big), \quad p_j^{(i)} \in \mathbb{R}^2, \tag{2}$$

$$\hat{y}_i^{\text{obj}} = \text{MLP}_{\text{obj}}\big(q_i'^{\text{obj}}\big) = (x_t, y_t, w, l, \theta_t, v_t^x, v_t^y), \tag{3}$$

where $q_i'$ is the enhanced instance-level query, $\hat{y}_i^{\text{map}}$ and $\hat{y}_i^{\text{obj}}$ are decoded information of map element and dynamic object. The former are represented as polyline, where $p_j^{(i)}$ is the 2D coordinates of the $j$-th point of map element, and $c^{(i)}$ denotes its class score. The latter includes the 2D coordinates of the object's center, its width and length, the heading angle, and the velocity components along the $x$ and $y$ axes at time $t$.

Next, instead of adopting the cascaded prediction–decision architecture, a unified decoder for both ego planning and object motion prediction is employed. Specifically, a learnable ego query $q_{\text{ego}}$ is augmented with a goal point embedding and an intent embedding corresponding to the high-level command $c$. For other agents, each motion query is formed by combining the updated object query $q_i'^{\text{obj}}$ with one of six implicit intent embeddings $e$. The two sets of queries are concatenated to form the agent-level motion query for the current scenario:

$$Q_{\text{motion}} = \text{Concat}\left(\text{MLP}_{\text{ego}}(q_{\text{ego}} \oplus g \oplus c), \text{MLP}_{\text{obj}}\left(\{q_{\text{obj}}'^{(i)} \oplus e_k, k = 1 \ldots K, i = 1 \ldots N_{\text{obj}}\}\right)\right) \tag{4}$$

In the unified decoder, agent-level motion queries attend to each other via self-attention and to map queries via cross-attention, enabling the model to capture interactions between agents constrained by the current road topology. The refined motion queries are again split into those for the ego and dynamic objects, which passes through seperated MLP for predicting the future trajectories. The planning and motion prediction process can be formally expressed as follows:

$$Q_{\text{motion}}' = \text{MotionDecoder}(q = Q_{\text{motion}}, kv = Q_{\text{map}}') \tag{5}$$

$$\hat{m}_j^k = \sigma\left(\text{MLP}_{\text{mod}}\left(q_{\text{m\_o}}^{j,k}\right)\right), \ \hat{\text{Traj}}_{obj}^{j,k} = \text{MLP}_{\text{traj\_o}}\left(q_{\text{m\_o}}^{j,k}\right), \ \text{where } k = 1,\dots,K, \ j = 1,\dots,N_{\text{obj}}, \ q_{\text{m\_o}}^{j,k} \in Q'_{\text{motion\_obj}} \tag{6}$$

$$\hat{\text{Traj}}_{\text{ego}} = \text{MLP}_{\text{traj\_e}}(q_{\text{m\_e}}), \quad q_{\text{m\_e}} = Q'_{\text{motion\_ego}} \tag{7}$$

where $\sigma$ denotes the sigmoid function; $\hat{m}_j^k$ represents the score of the $k$-th predicted modality for the $j$-th object, and $\hat{\text{Traj}}_{obj}^{j,k}$ denotes the corresponding trajectory over the prediction horizon $T$; $\hat{\text{Traj}}_{\text{ego}}$ is the planned trajectory of the ego vehicle.

In training phase, the loss terms of perception are same as in VAD [14], with imitation loss of L1 norm:

$$L_{\text{total}} = w_{\text{det\_map}}L_{\text{det\_map}} + w_{\text{det\_obj}}L_{\text{det\_obj}} + w_{\text{mod\_cls}}L_{\text{mod\_cls}} + w_{\text{motion\_obj}}L_{\text{motion\_obj}} + w_{\text{imi}}L_{\text{imi}} \tag{8}$$

The outputs of the perception and motion prediction tasks—namely, the implicit high-dimensional features and the structured trajectories and polylines—constitute the foundation of our self-supervised paradigm.

## 3.2 Multi-Modal Trajectory Planning Self-Supervision (MTPS) as Target Lane Selection

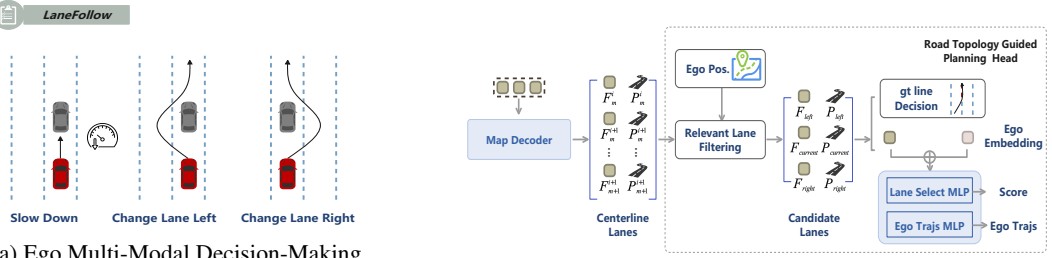

(a) Ego Multi-Modal Decision-Making in Cruising Scenarios (Lane Follow)

(b) Road Topology Guided Planning Head

Figure 2: Environment-Aware Lane Command and Road Topology Guided Planning.

Lane status in the surrounding environment plays a crucial role in guiding the ego's driving behavior. It defines the drivable area and constrains the range of feasible trajectories. As shown in Figure 2a, when the command is "LaneFollow" but an obstacle appears ahead, the ego can either decelerate in the current lane or change lanes to overtake. This turns lateral multi-modal planning into a lane selection problem shaped by the environment. This insight forms the basis of our Multi-Modal Trajectory Planning Self-Supervision(MTPS).

MTPS leverages the surrounding lane structure to guide the selection of the ego vehicle's planning modality. As illustrated in Figure 2b, this module includes a Road Topology Guided Planning Head, which generates both multi-modal decisions and corresponding ego trajectories, alongside a topology-aligned self-supervision mechanism. Firstly, a geometry-based lane filter is leveraged to select ego-relevant lane centerlines from the perception output. Given the set of lane centerline $P = \{P_j\}_{j=1}^{N_{map}}$, with each $P_j$ represented as a sequence of centerline points, we compute the minimum Euclidean distance $d_j$ and the relative angle $\varphi_j$ between the ego vehicle's current position $x_{\text{ego}}$, heading $\theta_{\text{ego}}$ and each centerline. The relevant lane set $FP = \{(F_{\text{left}}, P_{\text{left}}), (F_{\text{current}}, P_{\text{current}}), (F_{\text{right}}, P_{\text{right}})\}$ is constructed according to the following criteria, where $F$ denotes the implicit feature of each lane:

$$\begin{cases} d_j = \min\limits_{p \in P_j} \|x_{\text{ego}} - p\|_2 \\ \varphi_j = (p_j^* - x_{\text{ego}}) \times \begin{bmatrix} \cos\theta_{\text{ego}} \\ \sin\theta_{\text{ego}} \end{bmatrix} \end{cases}, \quad \begin{cases} (F_{\text{c}}, P_{\text{c}}) = (F_j, P_j), & \text{if } \exists d_j \leq 0.5W \\ (F_{\text{l}}, P_{\text{l}}) = (F_j, P_j), & \text{if } \exists d_j \in (0.5W, 1.5W] \text{ and } \varphi_j < 0 \\ (F_{\text{r}}, P_{\text{r}}) = (F_j, P_j), & \text{if } \exists d_j \in (0.5W, 1.5W] \text{ and } \varphi_j > 0 \\ (F, P) = (\mathbf{0}, \mathbf{0}), & \text{otherwise} \end{cases}$$
$$\tag{9}$$

where $p_j^*$ is the nearest point in $P_j$ to the ego vehicle, $W$ is the standard lane width, and the subscripts $c$, $l$, and $r$ denote the current, left, and right. If no centerline satisfies the above criteria, the corresponding feature $F$ and point set $P$ are set to zero.

This geometry-based filter is simple yet effective, allowing robust and efficient identification of the ego vehicle's current lane and adjacent lanes, thereby capturing all feasible lateral motion options.

Next, the implicit features of relevant centerlines are fused with the ego motion query. Two additional MLPs are utilized to predict the lane selection score and the corresponding trajectory. These scores are normalized by softmax operator, transforming the ego's multi-modal trajectory planning into a lane-level classification task, as described below:

$$H_i = q \oplus F_i, \ \forall F_i \in FP, \quad \boldsymbol{S} = \text{softmax}(\text{MLP}_{\text{score}}(\boldsymbol{H})), \quad \hat{\text{Traj}}_{\text{ego}} = \text{MLP}_{\text{traj}}\big(H_{\text{argmax}(\boldsymbol{S})}\big) \tag{10}$$

During training, the index of target centerline is provided by measuring the average spatial distance between the terminal portion of the ground-truth ego trajectory and each candidate lane centerline. The index of the centerline with the minimal distance is designated as the target lane index $l^*$, with the corresponding feature and polyline as $F^*$ and $P^*$. The loss function of selecting target lane is defined as:

$$L_{\text{MTPS}} = L_{\text{CE}}(\boldsymbol{S}, l^*) \tag{11}$$

### 3.3 Spatial Trajectory Planning Self-Supervision (STPS) based on Lane Centerline

Lane centerlines, compared to other road topology cues like markings and boundaries, play a more critical role in learning robust driving behaviors. Human trajectories often deviate slightly from the centerline due to factors like driving style, weather, or control noise—difficult to attribute and thus regarded by the model as learning noise. This noisy supervision can negatively impact the model's causal understanding of driving behaviors, particularly in scenarios involving intersection turning. Lane centerlines naturally connect incoming and outgoing lanes, and training on trajectories that deviate from them increases the risk of drifting into the wrong lane due to cumulative errors.

Building on this insight, we propose a Spatial Trajectory Planning Self-Supervision (STPS) mechanism, in which the expert trajectory $Traj_{gt}$ is replaced by a centerline-aligned version as the primary supervision signal. Specifically, each expert trajectory point is checked against nearby target centerline points (from the previous stage); if a matched centerline point is found, it replaces the original point. Original expert point is retained only when no point is matched, which serves as a regularization term to preserve the smoothness of the target trajectory. The resulting trajectory $Traj'_{tgt}$ supervises the Road topology guided trajectory regression head, working as a spatial ground-truth path—free of temporal bias but more causally aligned. Formally, for expert trajectory point $Traj_{gt}^t$ at time step $t$, the resulting updated trajectory points $Traj'^t_{tgt}$ are given by:

$$p'_t = \arg \min_{p'_j \in P^*} \|Traj_{gt}^t - p'_j\|_2 \tag{12}$$

$$Traj'^t_{tgt} = \begin{cases} p'_t, & \text{if } \|Traj_{gt}^t - p'_t\|_2 \leq w \\ Traj_{gt}^t, & \text{otherwise} \end{cases} \tag{13}$$

This new trajectory $Traj'_{tgt}$ is then used to supervise the trajectory regression head as:

$$L_{\text{STPS}} = \frac{1}{N} \sum_{t=1}^{N} \|\hat{Traj}_{ego}^t - Traj'^t_{tgt}\|_1 \tag{14}$$

Last but not least, since the regression head also takes the target centerline features $F^*$ as input, aligning the trajectory target with the centerline further strengthens causal reasoning in trajectory prediction by jointly leveraging topological cues and supervision consistency.

### 3.4 Negative Planning Self-supervision (NPS) from Dynamic Objects' Future Bounding Boxes

An autonomous systems must dynamically respond to surrounding agents. While MTPS and STPS support positive causal modeling for general planning, safe interaction requires negative causal modeling to proactively avoid risky outcomes—e.g., adjusting the ego trajectory to prevent overlap with the predicted motion of an encroaching parked vehicle as shown in the Figure 3a.

Motivated by this insight, we propose the Negative Trajectory Planning Self-Supervision (NTPS) mechanism, which imposes safety constraints on ego planning using the predicted future bounding boxes of surrounding agents. As illustrated in Figure 3b, we construct future bounding box sequences for both ego and dynamic objects using predicted trajectories and perceived object dimensions. The

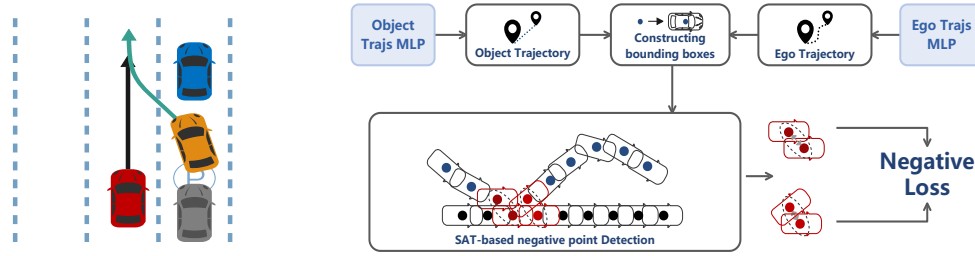

(a) Example of Risky Interaction Scenario      (b) Safety Constraint Mechanism of NTPS

Figure 3: Negative Trajectory Planning Self-Supervision (NTPS) for Safety-Constrained Ego.

orientation at each timestep is estimated via trajectory offsets, and overlap detection is performed using the Separating Axis Theorem (SAT) [9]. Upon detecting overlaps, we introduce a negative supervision loss that penalizes ego trajectories encroaching into occupied space by encouraging divergence from the overlapping region. as follows:

$$L_{\text{NTPS}} = \sum_{t \in T_{\text{coll}}} \max(0, \beta - \|\hat{Traj}^t_{\text{ego}} - \hat{Traj}^t_{\text{obj\_col}}\|_2) \qquad (15)$$

where $t \in T_{\text{coll}}$ denotes each timestep in the set of detected collision timesteps $T_{\text{coll}}$, and $\hat{Traj}^t_{\text{obj\_col}}$ represents the trajectory point of the surrounding object that is predicted to collide with the ego vehicle at timestep $t$.

In this process, SAT-based overlap detection identifies risk-inducing points along the trajectories of surrounding agents. These are treated as negative supervision signals, guiding the ego trajectory to diverge from potential collision zones and thereby enhancing safety in dynamic interactions.

### 3.5 Perception-Guided Self-supervision in Optimization

During training, the proposed PGS paradigm introduces guidance from perception by integrating the three distinct self-supervision losses described above to total loss introduced in Section 3.1 as:

$$L'_{\text{total}} = L_{\text{total}} + w_{\text{MTPS}} L_{\text{MTPS}} + w_{\text{STPS}} L_{\text{STPS}} + w_{\text{NTPS}} L_{\text{NPS}} \qquad (16)$$

where $w$ represent the relative importance of each loss component.

## 4 Experiment

### 4.1 Dataset & Metrics

**Dataset:** To evaluate the real-world effectiveness of our self-training paradigm, we evaluate on the challenging closed-loop benchmark Bench2Drive [10], built on CARLA v2. The dataset includes 1,000 short clips across 44 complex scenarios (950 for training, 50 for open-loop validation) and 220 predefined routes for standardized closed-loop evaluation nd fair performance comparison.

**Metrics:** We adopt Bench2Drive's official metrics: Driving Score, Success Rate, Efficiency, and Comfortness for closed-loop evaluation, and L2 Displacement Error (L2) for open-loop evaluation.

### 4.2 Implementation Details

The training process of PGS is divided into two stages, each with distinct learning objectives.

Stage 1 focuses on perception learning. We enhance the online map detection task by introducing lane centerlines as a new class of map elements. While the task of motion prediction of dynamic objects is trained in this phase as well. In addition, traffic light detection from front-view images is incorporated to capture critical causal dependencies for safely navigating signalized intersections. Training in this stage lasts for 6 epochs.

Stage 2 builds upon the perception capabilities from Stage 1 and introduces joint optimization of the perception module and self-supervised objectives. Perception losses are retained to maintain accurate environmental understanding, while three self-supervised losses—MTPS, STPS, and NTPS—are introduced to supervise ego planning tasks. Stage 2 is also trained for 6 epochs.

Training is conducted on 16 NVIDIA RTX V100 GPUs using the AdamW [21] optimizer, with a weight decay of 0.01 and an initial learning rate of 4e-4. The loss weights are set to $w_{\text{MTPS}} = 1.0$, $w_{\text{STPS}} = 0.3$, and $w_{\text{NPS}} = 1.0$.

## 4.3 Comparison with State-of-the-Art Methods

Table 1: Open-loop and Closed-loop results of planning in Bench2Drive. Avg. L2 is averaged over the predictions in 2 seconds under 2Hz. * denotes expert feature distillation.

| Method | Avg. L2 ↓ | Driving Score ↑ | Success Rate (%) ↑ | Efficiency ↑ | Comfortness ↑ |
|---|---|---|---|---|---|
| AD-MLP [28] | 3.64 | 18.05 | 0.00 | 48.45 | 22.63 |
| UniAD-Base [8] | 0.73 | 45.81 | 16.36 | 129.21 | 43.58 |
| UniAD-Tiny [8] | 0.80 | 40.73 | 13.18 | 123.92 | 47.04 |
| VAD-Base [14] | 0.91 | 42.35 | 15.00 | 157.94 | 46.01 |
| VAD-Tiny [14] | 1.15 | 34.28 | 10.45 | 70.04 | **66.86** |
| SparseDrive [23] | 0.87 | 44.54 | 16.71 | 170.21 | 48.63 |
| GenAD [29] | - | 44.81 | 15.90 | - | - |
| DiFSD [22] | 0.70 | 52.02 | 21.00 | 178.30 | - |
| DriveTransformer [13] | **0.62** | 63.46 | 35.01 | 100.64 | 20.78 |
| DiffAD [24] | - | 67.92 | 38.64 | - | - |
| WoTE [16] | - | 61.71 | 31.36 | - | - |
| BridgeAD | 0.71 | 50.06 | 22.73 | - | - |
| **PGS(Ours)** | 0.77 | **78.08** | **48.64** | **181.31** | 12.37 |
| TCP-traj* [27] | 1.70 | 59.90 | 30.00 | 76.54 | 18.08 |
| ThinkTwice* [12] | 0.95 | 62.44 | 31.23 | 69.33 | 16.22 |
| DriveAdapter* [11] | 1.01 | 64.22 | 33.08 | 70.22 | 16.01 |

Table 1 summarizes the comparative open-loop and closed-loop planning performance on Bench2Drive. Compared to VAD-Base—the baseline model for our approach—PGS reduces the open-loop L2 error from 0.91 to 0.77. more importantly, PGS achieves a remarkable Driving Score of 78.08, outperforming VAD-Base (42.35) by 35.73 points in closed-loop evaluation. The Success Rate improves significantly from 15.00% to 48.64%. These improvements are primarily attributed to the enhanced causal reasoning capabilities introduced by our self-supervised planning framework.

The comparison with contemporaneous methods [13, 24, 16] further validates the effectiveness of our proposed paradigm. These methods improve closed-loop performance by adopting more complex architectures, leveraging multi-modal sensor inputs, employing diffusion models for multi-modal decision, or combining trajectory generation with online ranking strategies, but still primarily rely on imitation learning and lack explicit consideration of causal reasoning. In contrast, the self-supervised, causality-driven PGS framework consistently outperforms these methods, highlighting the effectiveness of perception-guided self-supervision in capturing causal relationships.

Furthermore, PGS surpasses methods based on knowledge distillation (e.g., [27, 12, 11]) as well. Although distillation enhances planner robustness by transferring knowledge from expert models trained with noise-free privileged information, it fails to model the causal dependencies between redundant perception outputs and ego planning, and results in causal confusion and suboptimal decision-making policies. These results further validate the superiority of PGS in achieving causally grounded and robust driving performance.

Table 2 further compares the success rates of different approaches across specific driving scenarios. A scenario is considered successful only if the ego vehicle reaches the designated destination without any collisions or infractions. Our model consistently outperforms competitors across several critical driving skills, achieving notably high success rates in Merging (35.00%), Overtaking (73.33%), Emergency Braking (55.00%), and Give Way (60.00%). It also obtains the highest overall ability score of 53.40%. These results highlight the strong generalization capability of our approach in handling complex and highly interactive scenarios.

Table 2: **Multi-Ability Results of E2E-AD Methods.** * denotes expert feature distillation.

| Method | Ability (%) ↑ | | | | | |
| --- | --- | --- | --- | --- | --- | --- |
| | Merging | Overtaking | Emergency Brake | Give Way | Traffic Sign | **Mean** |
| AD-MLP [28] | 0.00 | 0.00 | 0.00 | 0.00 | 4.35 | 0.87 |
| UniAD-Tiny [8] | 8.89 | 9.33 | 20.00 | 20.00 | 15.43 | 14.73 |
| UniAD-Base [8] | 14.10 | 17.78 | 21.67 | 10.00 | 14.21 | 15.55 |
| VAD [14] | 8.11 | 24.44 | 18.64 | 20.00 | 19.15 | 18.07 |
| DriveTransformer [13] | 17.57 | 35.00 | 48.36 | 40.00 | 52.10 | 38.60 |
| DiffAD [24] | 30.00 | 35.55 | 46.66 | 40.00 | 46.32 | 38.79 |
| **PGS (Ours)** | **35.00** | **73.33** | **55.00** | **60.00** | 43.68 | **53.40** |
| TCP* [27] | 16.18 | 20.00 | 20.00 | 10.00 | 6.99 | 14.63 |
| TCP-ctrl* | 10.29 | 4.44 | 10.00 | 10.00 | 6.45 | 8.23 |
| TCP-traj* | 8.89 | 24.29 | 51.67 | 40.00 | 46.28 | 34.22 |
| TCP-traj w/o distillation | 17.14 | 6.67 | 40.00 | 50.00 | 28.72 | 28.51 |
| ThinkTwice* [12] | 27.38 | 18.42 | 35.82 | 50.00 | 54.23 | 37.17 |
| DriveAdapter* [11] | 28.82 | 26.38 | 48.76 | 50.00 | **56.43** | 42.08 |

## 4.4 Ablation Study on Bench2Drive

We conduct extensive ablation experiments to assess the contribution of each component in our self-supervised paradigm. For efficient closed-loop evaluation, we select the **Merging** and **Overtaking** scenarios, which are both complex and highly interactive. Together, they account for more than half of the total scenarios, making the evaluation metrics on them sufficiently representative.

Table 3: Ablation Study of the Proposed PGS Framework.

| Method | Avg. L2 | Ability (%) ↑ | | |
| --- | --- | --- | --- | --- |
| | | Merging | Overtaking | **Mean** |
| VAD-Base | 0.91 | 8.11 | 24.44 | 16.28 |
| VAD-Tiny | 1.15 | 9.33 | 11.11 | 10.22 |
| $PGS_{Base}$ | 0.87 | 16.46 | 13.33 | 14.89 |
| $PGS_{Base+STPS}$ | 0.78 | 24.44 | 26.25 | 25.35 |
| $PGS_{Base+STPS+MTPS}$ | 0.75 | 23.75 | 44.44 | 34.10 |
| $PGS_{All}$ | 0.77 | 35.00 | 73.33 | 54.17 |
| $PGS_{NTPS}$ | 0.90 | 25.00 | 6.67 | 15.84 |
| $PGS_{self}$ | 2.89 | 31.25 | 35.56 | 33.40 |

As shown in Table 3, $PGS_{Base}$ denotes our baseline model, where the perception network is trained with the perception loss used in VAD, and the planning head is trained with the imitation loss. Compared to VAD, it achieves a slightly lower L2 error and a comparable success rate. $PGS_{STPS}$ introduces the centerline-aligned Spatial Trajectory Planning Self-Supervision, which strengthens the alignment between road topology cues and the ego's planned trajectory, leading to significant improvements in both L2 error and success rate. Building upon this, $PGS_{STPS+MTPS}$ incorporates a relevant lane filter and reformulates the multi-modal ego decision as a lane selection task within this filtered set. This design yields a substantial performance boost in the **Overtaking** scenarios, where frequent lane changes occur. Finally, $PGS_{All}$ further adds Negative Trajectory Planning Self-Supervision by identifying risky future positions of surrounding dynamic objects and penalizing ego trajectories that overlap with them. This additional constraint reduces collision risk in both selected scenarios. Overall, $PGS_{All}$ achieves a mean success rate of 54.17%, with an improvement of over 39% compared to $PGS_{Base}$, which is trained purely via imitation learning. We further isolate the effect of Negative Trajectory Planning Self-Supervision through a dedicated variant, $PGS_{NTPS}$, to assess model behavior in unstructured environments. Despite the absence of structured road priors, this variant exhibits strong performance in Merging scenarios, demonstrating the capability of NTPS to mitigate collision risks in complex, geometry-agnostic contexts. However, its performance degrades in Overtaking scenarios, likely due to overly conservative behavior in the presence of static obstacles and the lack of contextual lane information. These limitations are effectively addressed

when NTPS is integrated with STPS and MTPS, as structured priors offer richer spatial cues and broaden the maneuver space, enabling more balanced and flexible decision-making.

Besides, we retrained a model, $PGS_{self}$, using only PGS self-supervision, without any imitation loss. As expected, the L2 error increases significantly due to the absence of expert trajectory knowledge. However, it still achieves a respectable success rate of 33.40%, outperforming both $PGS_{Base}$ and $VAD$ by a large margin. This underscores the importance of perception-consistent ego planning in causal modeling.

## 5    Conclusion & Limitation

**Conclusion:**  We introduce a perception-guided self-supervision paradigm for end-to-end autonomous driving. By leveraging road topology and dynamic agent motion as both inputs and supervisory signals, our approach aligns ego trajectory prediction with structured, causally relevant cues, enabling stronger causal reasoning and state-of-the-art closed-loop performance. Extensive ablation studies further substantiate the efficacy of our self-supervision mechanisms, highlighting a promising direction for enhancing the real-world robustness of end-to-end autonomous driving.

**Limitation:** The framework's success relies on accurate and robust perception of dynamic/static agents and road structures. Limited perception accuracy or generalization may impair planning performance, making perception robustness a key challenge for future work.

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

# A    Qualitative Analysis of Lane Centerline Perception

In CARLA, road topology information is stored using a graph structure, where each node represents an individual lane. The attributes of each node include the global coordinates of the lane's centerline, lane width, and other geometric properties. For each frame of training data, the ground-truth centerlines near the ego vehicle are generated by transforming all lane centerline coordinates from the global map into the ego-centric local coordinate system based on the recorded ego pose.

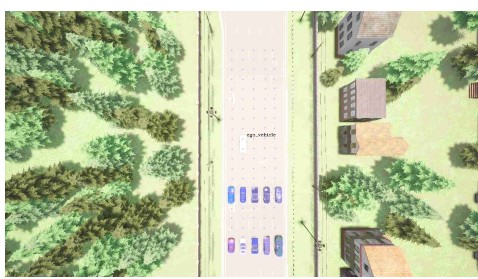

(a) Perception results in straight road scenarios

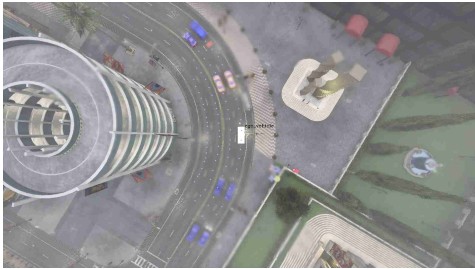

(b) Perception results in curved road scenarios

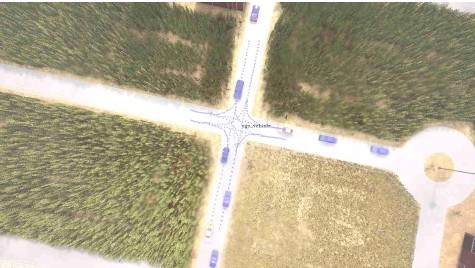

(c) Perception results at intersections

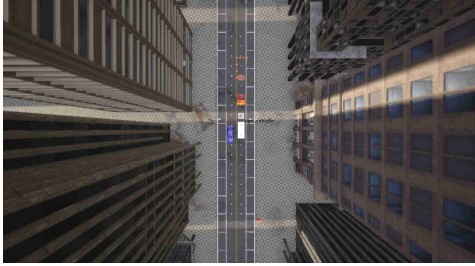

(d) Perception results in obstacle-present scenarios

Figure 4: Visualization of lane centerline perception under diverse road conditions

The qualitative examples presented in Figure 4 demonstrate that, even in the absence of salient visual features for lane centerlines, the map perception module in $PGS$ can still reliably detect them by leveraging visual context from the scene. This ability is particularly evident in complex areas such as intersections. Such robust perception of centerlines forms the foundation for the effectiveness of the self-supervised MTPS and STPS components in our framework.

# B    Relevant Lane Filtering and Target Lane Determination in MTPS

In the Multi-Modal Trajectory Planning Self-Supervision (MTPS), we frame the ego vehicle's multi-modal decision-making as a target lane selection task. The relevant lane filter extracts candidate lanes—namely, the left, current, and right lanes—based on the perceived road topology. Then, by referencing the ground-truth trajectory, the system identifies the target lane corresponding to the intended driving behavior.

**Visualization of Supervision Signal Construction for Target Lane Selection**

This subsection visually illustrates how the Multi-modal Trajectory Planning (MTP) module leverages the surrounding lane topology to inform the selection of the ego vehicle's planning modality, as described in Section 3.2. Figure 5 provides a step-by-step illustration of the pipeline for constructing the supervision signal for target lane selection, starting from the perception outputs, moving through the filtering of relevant candidate lanes, and culminating in the determination of the target lane by referencing the endpoint of the ground-truth trajectory.

**Classification Accuracy of the Target Lane Selection Task**

To assess the effectiveness of the topology-guided supervision in MTPS, we evaluate the classification accuracy and recall of target lane prediction on the B2D open-loop validation set. As shown in

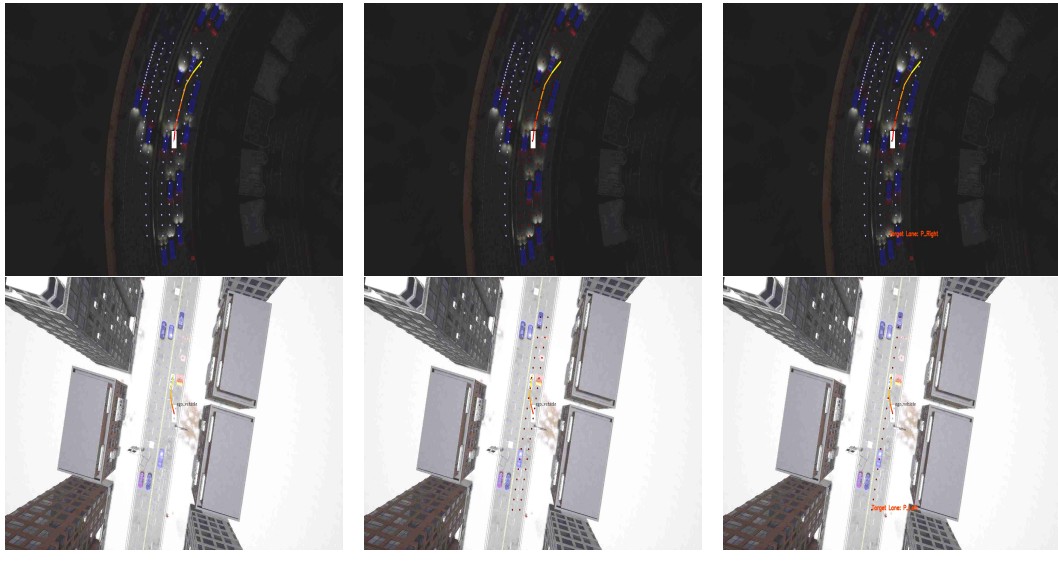

| (a) Perceived lane centerlines | (b) Selected relevant lane set | (c) Selected target lane |

Figure 5: Visualization of Target Lane Selection. Light pink points indicate the perceived lane centerlines, while dark red points represent the selected relevant lane set. The final target lane is also highlighted in dark red, with its name labeled in orange.

Table 4, training with the full 3-second ground-truth trajectory and standard cross-entropy (CE) loss achieves high accuracy across all classes. However, due to severe class imbalance, the recall for rare categories such as "oriented to left lane" and "oriented to right lane" remains relatively low, at 0.63 and 0.80 respectively.

Table 4: Accuracy and Recall of Target Lane Classification under Different Training Strategies.

| Metric | Label | CE | Weighted CE | Weighted CE + 2s Trajectory |
|---|---|---|---|---|
| Accuracy | Oriented to current lane | 0.9685 | 0.9554 | 0.9690 |
| | Oriented to left lane | 0.9785 | 0.9669 | 0.9748 |
| | Oriented to right lane | 0.9887 | 0.9832 | 0.9909 |
| Recall | Oriented to current lane | 0.9855 | 0.9586 | 0.9676 |
| | Oriented to left lane | 0.6268 | 0.7344 | 0.8788 |
| | Oriented to right lane | 0.7978 | 0.9011 | 0.9233 |

To mitigate this imbalance, we adopt a class reweighting strategy based on inverse class frequencies. Specifically, class weights are computed from the training data distribution as [1.074, 32.480, 26.505], corresponding to current, left, and right lane orientations, respectively. This adjustment significantly improves recall for the left and right lane categories by approximately 10 percentage points.

Furthermore, when shortening the trajectory horizon used for matching from 3 seconds to 2 seconds, both recall and precision improve. Shorter trajectories more accurately reflect immediate driving intentions, reducing ambiguity. Based on these results, we adopt inverse frequency weighting and 2-second trajectory matching as the default configuration for PGS training in MTPS.

## C   Spatial Trajectory Generation in STPS

In STPS, we construct a spatial trajectory by combining the target lane centerline with the ground-truth trajectory, which is then used as supervision for the trajectory regression head to facilitate better causal modeling.

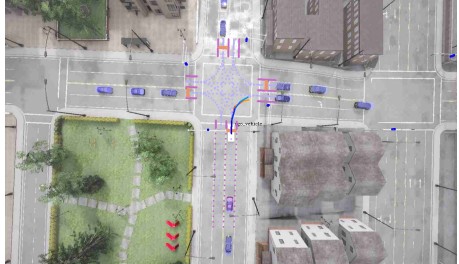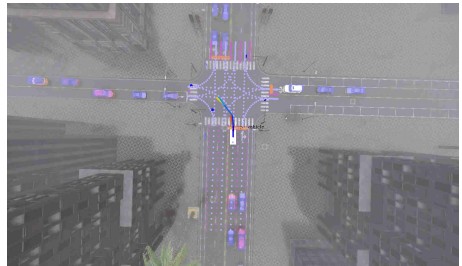

(a) Ground-truth and STP trajectories in turning scenarios at intersections

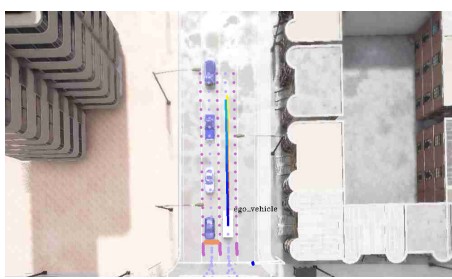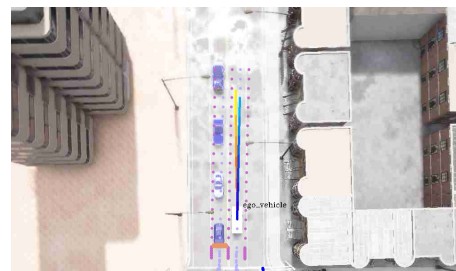

(b) Ground-truth and STP trajectories in straight-driving scenarios

Figure 6: Comparison of STP-generated and ground-truth trajectories. Perceived lane centerlines are rendered as light pink points, and other perceived lanes in orchid. Ground-truth trajectories are visualized using red-to-yellow gradients, while STP trajectories are represented by blue gradients.

As illustrated in Figure 6, we present two representative scenarios. In Figure 6a, when navigating intersections, ground-truth trajectories occasionally deviate from the centerline of the designated outbound lane—either drifting left or right. Such sporadic deviations, likely due to labeling noise or imperfect execution, introduce spurious signals that hinder the model's ability to capture causally valid motion patterns. In contrast, STP trajectories, aligned with the intended lane centerline, exhibit high directional consistency and semantic alignment, thereby facilitating more robust causal learning.

Similarly, in Figure 6b, ground-truth trajectories in straightforward driving scenarios exhibit oscillations around the lane centerline, potentially introducing ambiguity in lane-following behavior. The STP-generated trajectories, by contrast, maintain a stable, forward-directed course, offering clearer intent supervision and reducing trajectory-level noise.

## D    Separating Axis Theorem (SAT) Algorithm Description

In NTPS, the SAT[20] algorithm is employed to generate negative supervision signals for the ego vehicle trajectory, conditioned on the predicted trajectories of surrounding agents.

SAT is a classical method for collision detection between convex polygons in 2D space. It is based on the principle that two convex polygons do not intersect if and only if there exists a separating axis—an axis along which the projections of the two polygons do not overlap. In practice, the set of candidate axes is constructed by computing the outward normals of all edges from both polygons. If a separating axis is found, the polygons are guaranteed to be disjoint. Otherwise, the polygons must intersect.

As shown in Algorithm 1, the SAT algorithm iteratively tests all potential separating axes derived from the polygon edges.

**Algorithm 1:** Separating Axis Theorem (SAT)

---

**Input:** Vertex sets of polygons $A$ and $B$
**Output:** Whether they intersect (true/false)
**for** *each polygon $P \in \{A, B\}$* **do**
    **for** *each edge $e$ of $P$* **do**
        Compute edge normal as projection axis **axis**;
        Project polygons $A$ and $B$ onto **axis**, obtaining intervals $proj_A$ and $proj_B$;
        **if** $proj_A$ and $proj_B$ *do not overlap* **then**
            **return** false ;   // Separating axis found - polygons don't intersect

**return** true ;       // No separation found on any axis - polygons intersect

---

# E   Visualization of Closed-loop Evaluation

**Representative Cases in Diverse and Complex Scenarios**

Figure 7 presents the closed-loop evaluation results of our model, $PGS_{All}$, in the CARLA simulation environment. These visualizations are drawn from the full set of 220 closed-loop test scenarios in the Bench2Drive benchmark, encompassing a diverse spectrum of challenging conditions, such as adverse weather (e.g., heavy rain, dense fog), varied lighting (e.g., daytime, nighttime), and complex traffic scenes (e.g., intersections, lane merging, overtaking, and traffic light negotiation). As illustrated, the model consistently produces smooth, goal-directed trajectories that respect the intended global route while dynamically responding to contextual hazards.

The results indicate that $PGS_{All}$ is capable of dynamically adjusting its trajectory to avoid obstacles while maintaining route fidelity. The model exhibits a strong capacity to align its predictions with the underlying semantic road structure, reflecting a nuanced understanding of the causal relationships between environmental cues and appropriate driving behavior. This capability contributes to reliable and safe autonomous decision-making in closed-loop execution.

**Limitations and Failure Case Analysis**

While $PGS_{All}$ demonstrates robust performance across a wide range of scenarios, we observe several failure cases that expose current limitations in perception and causal reasoning.

As shown in Figure 8a, the ego vehicle fails to avoid a parked car with an opened door. This failure is attributed to the perception module treating such vehicles the same as regular static obstacles, without distinguishing the opened door as a separate semantic element. Consequently, the model fails to learn the causal relationship between door-opening events and the necessity of avoidance. Similarly, in Figure 8b, the model does not yield to an approaching emergency vehicle from behind, likely due to the absence of semantic differentiation between emergency and regular vehicles in the perception process.

These cases suggest the need to enhance the perception module by introducing specialized object categories or refining bounding box representations (e.g., to cover opened doors or siren-bearing vehicles). Such improvements would allow the model to better capture the causal structures required for socially compliant and context-aware planning in these critical situations.

**Supplementary Visualizations and Reproducibility**

To further support our analysis, we provide an extended set of visualizations in video format, accessible via the following GitHub repository: Supplementary Materials. Within this repository, the folder `0_representative_cases` contains two subfolders, `DiverseChallengingScenarios` and `FailureScenarios`, which showcase representative cases that highlight both the strengths and limitations of the proposed model.

Full closed-loop evaluation results are available in the folder `1_PGS_all_metric_files`, which contains `merged.json` (overall results) and `merged_ability.json` (per-scenario metrics). Frame-level logs for all 220 scenarios, including ego states, control commands (steering, throttle, brake), predicted traffic light states, and high-level decisions, are stored in `eval_PGS_all`.The evaluation

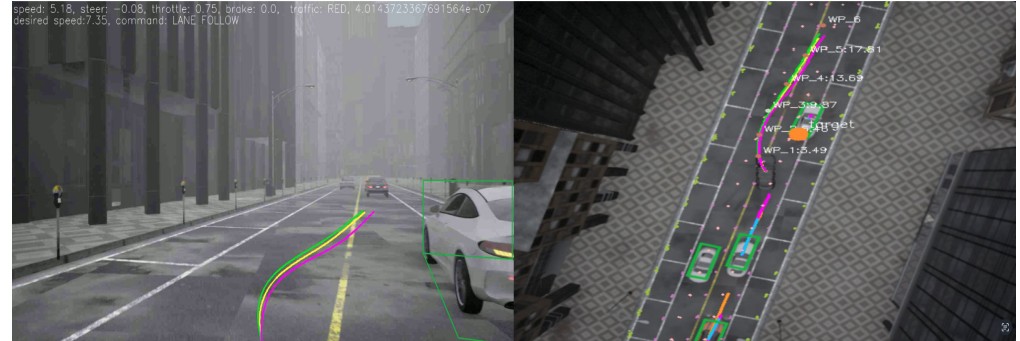

(a) Overtaking a stationary vehicle

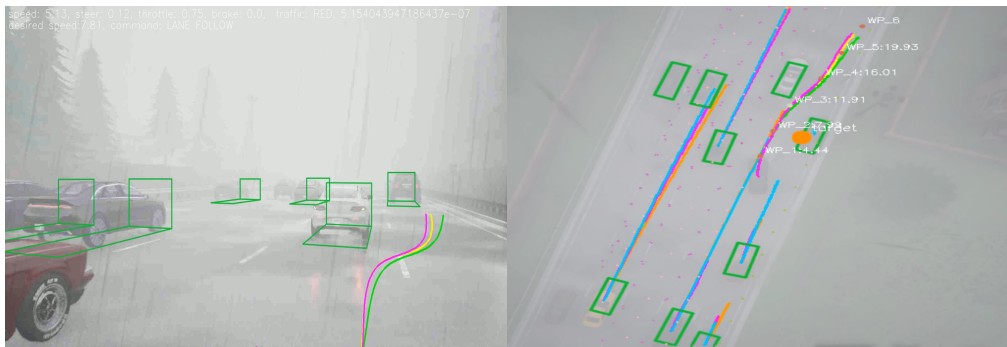

(b) Lane change under rainy weather conditions

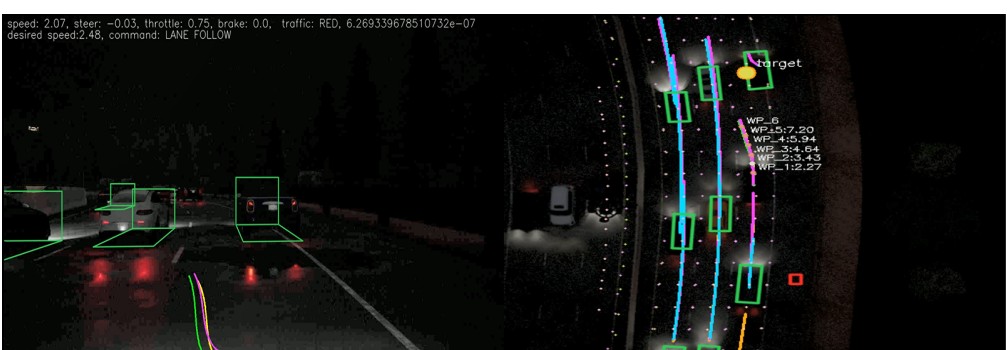

(c) Driving at night with low visibility

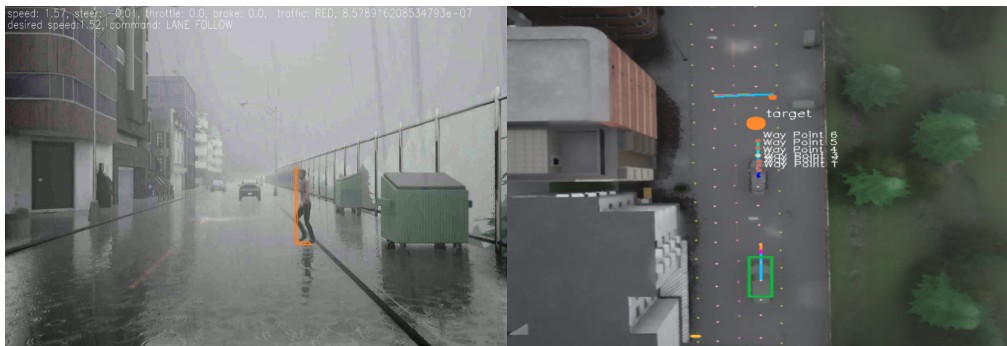

(d) Pedestrian avoidance in straight-road scenarios

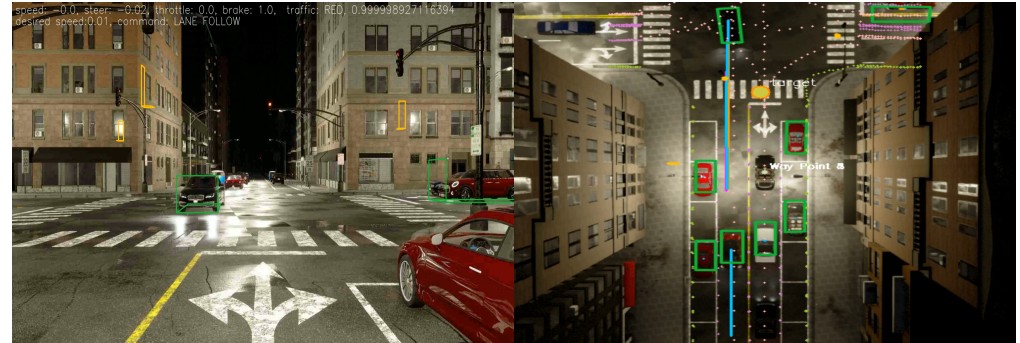

(e) Traffic light recognition in the evening

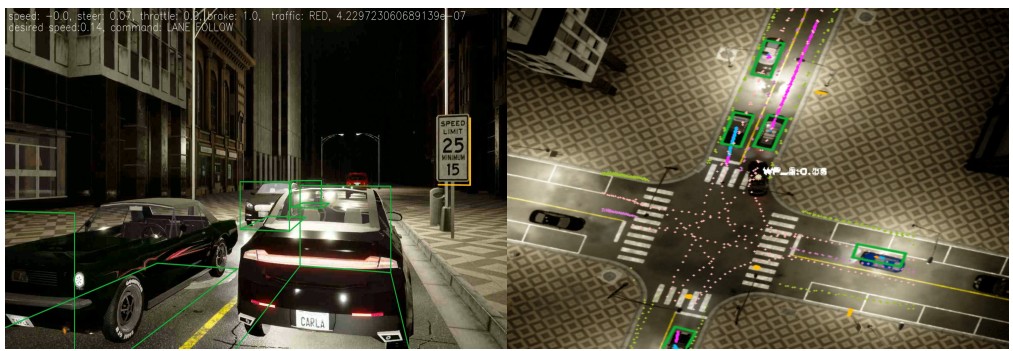

(f) Emergency stop due to malfunctioning vehicle ahead

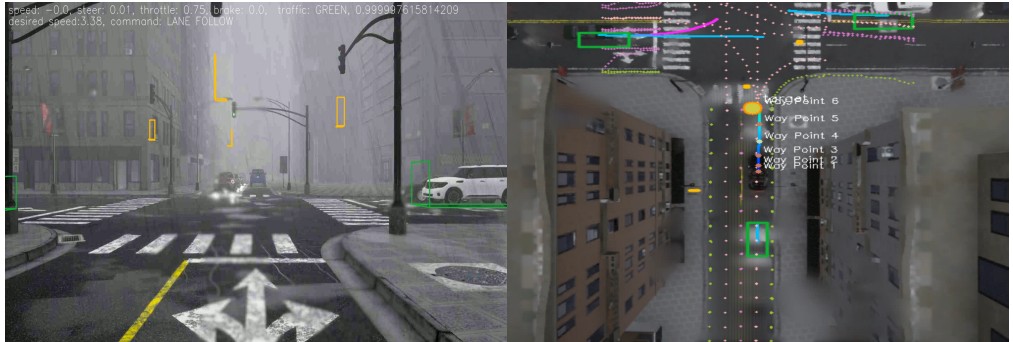

(g) Traffic light recognition under rainy weather conditions

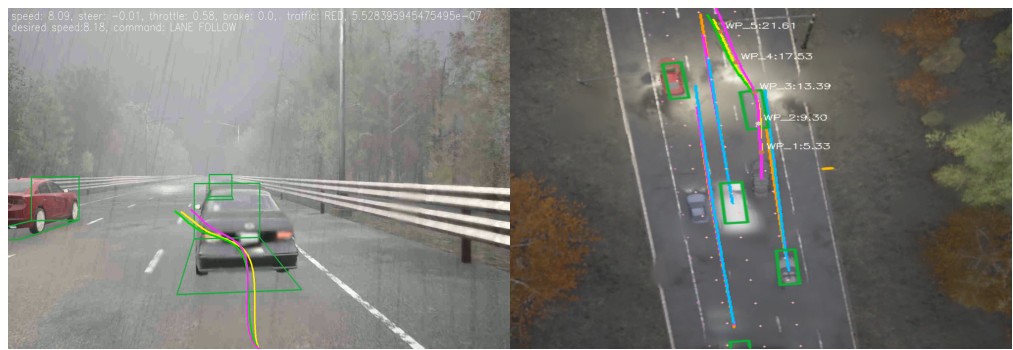

(h) Vehicle yielding right of way

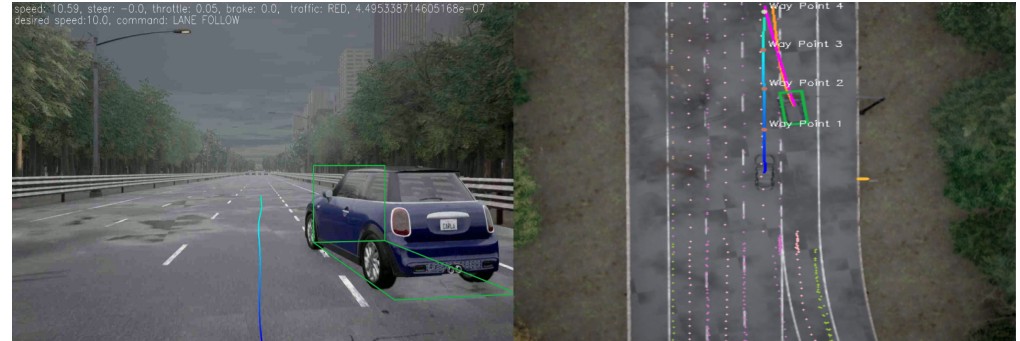

(i) Merging into highway with yielding

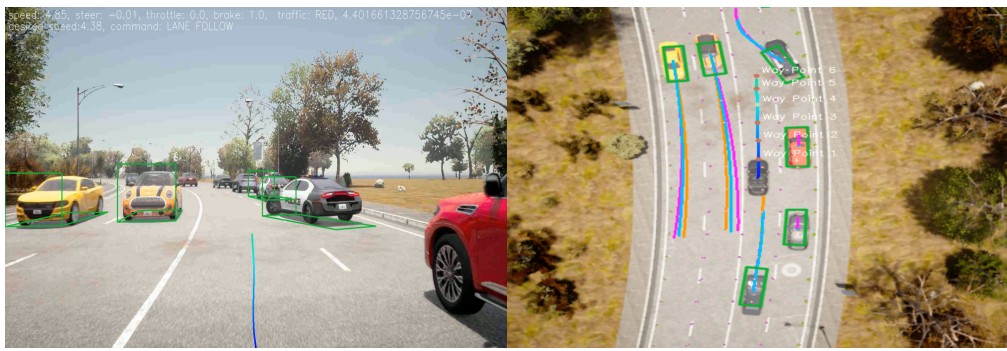

(j) Merging into highway with a parked vehicle ahead

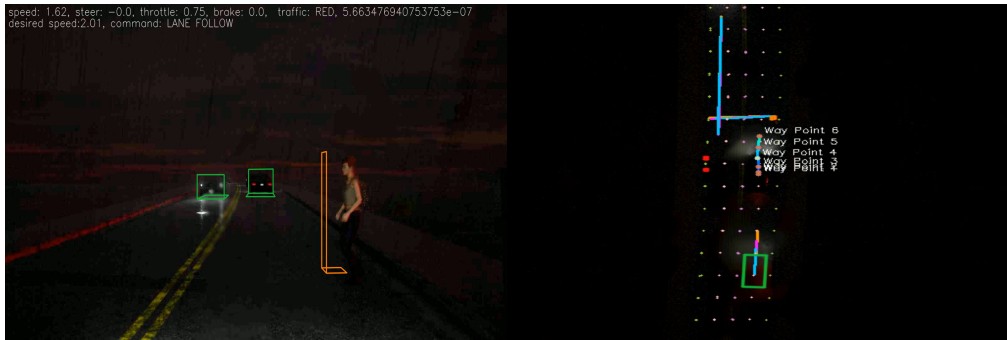

(k) Pedestrian violation crossing the road at night

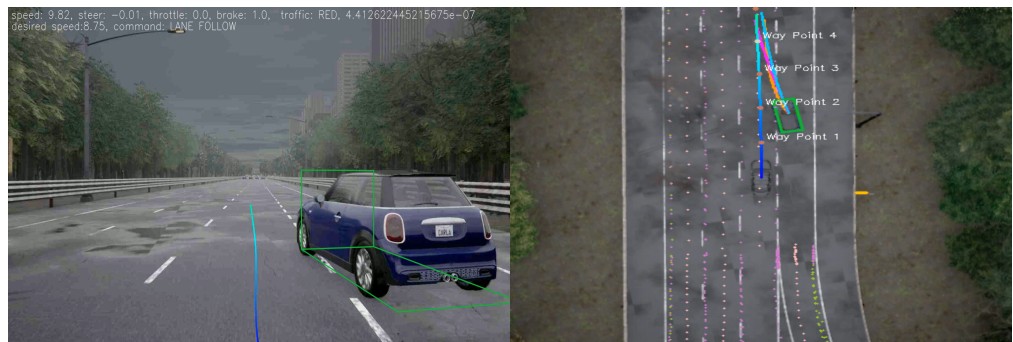

(l) Obstacle avoidance when other vehicles merge into the lane

Figure 7: Visualization of representative closed-loop scenarios from the Bench2Drive benchmark. The wp or way point denotes predicted trajectory points. The target is the mid-range goal issued by the global route planner in simulation environment, encoded as input to the PGS planning network.

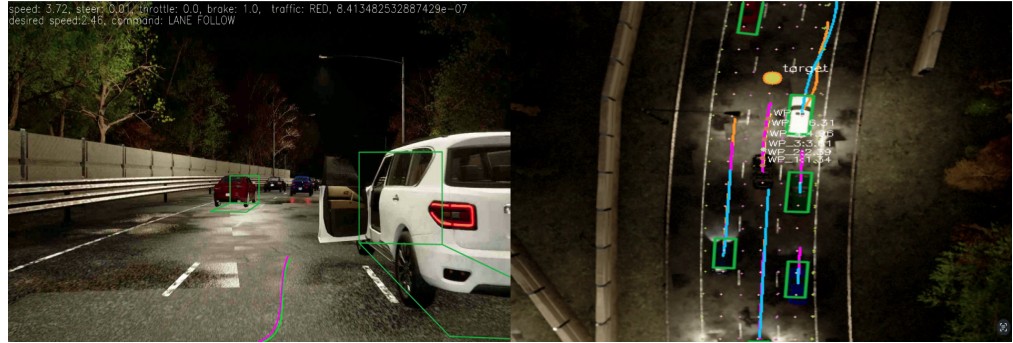

(a) Failure to avoid a vehicle with an opened door.

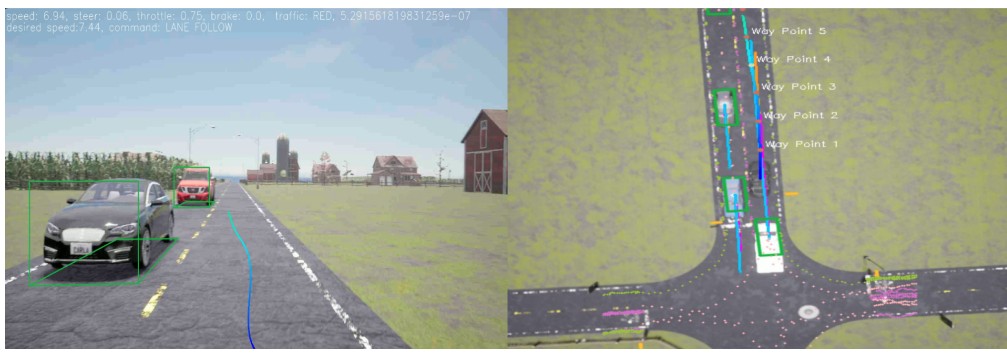

(b) Failure to yield to an approaching emergency vehicle.

Figure 8: Representative failure cases due to incomplete perception and missing causal cues.

process can be reproduced for each individual town by following the instructions in the Metric section of the Eval Tools.

## F   PID Controller Configuration

We build upon the original PID controller parameters from Bench2Drive[10] and modify the aim point selection strategy. Instead of using a fixed 4.0-meter target, we adopt a dynamic approach based on vehicle speed: selecting a near aim point of 4.0 meters for low-speed scenarios (below 6.5 m/s) and a far aim point of 10.0 meters for high-speed scenarios (above 6.5 m/s). This design stems from the principle that at higher speeds, the vehicle requires a longer look-ahead distance to follow the planned trajectory accurately. By providing sufficient foresight, the farther aim point facilitates smoother steering adjustments, thereby enhancing trajectory stability and reducing oscillations observed during high-speed maneuvers in our experiments.

## G   Experiments compute resources

All experiments were conducted with the following specifications:

**Hardware Configuration**

- **CPU:** Each node is equipped with dual *Intel(R) Xeon(R) Gold 6278C @ 2.60GHz* processors, providing 52 physical cores and 104 threads per node. With 2 nodes in total, the system utilizes 4 CPU sockets and 208 logical processors.

- **Memory:** 512 GB RAM per node

- **GPU:** 16 *NVIDIA V100* GPUs (32 GB each), with 8 GPUs per node across 2 nodes

**Training Time**

- **Stage 1 Training:** Completed in approximately 1 day using 16 GPUs, equivalent to around 384 GPU hours.

- **Stage 2 Training:** Completed in approximately 1 day using 16 GPUs, equivalent to around 384 GPU hours.

