# OpenReview forum: "Prioritizing Perception-Guided Self-Supervision: A New Paradigm for Causal Modeling in End-to-End Autonomous Driving"
_NeurIPS.cc/2025/Conference — NeurIPS 2025 poster_

### Official Review · Reviewer_gAQK · 2025-06-30

**Clarity:** 3
**Significance:** 3
**Originality:** 2
**Rating:** 4
**Confidence:** 4

**Summary:**

Conventional imitation-learning-based end-to-end autonomous driving systems achieve strong performance in open-loop evaluations by leveraging large-scale expert demonstrations, yet they suffer severe degradation in closed-loop (real-time) settings due to causal confusion. To address this issue, the authors introduce Perception-Guided Self-Supervision (PGS). PGS treats perception outputs, such as lane centerlines and predicted trajectories of nearby agents, as the primary supervisory signals for the decision-making module, and applies positive and negative self-supervision to align the ego vehicle’s trajectory with these perception cues, thereby alleviating causal confusion.

**Questions:**

1. The proposed PGS is applied only to the Ego Planning Head. Could it also be leveraged for the branches that output object trajectories and mode-classification scores?
2. What does $\beta$ represent in line 240?
3. Section 4.3 reports high driving-score and success-rate figures, yet noticeably lower scores for efficiency and comfort. What accounts for this imbalance?
4. Are “PPGSS” and “PGS” in Tables 1 and 2 the same? If not a typo, please clarify what PPGSS stands for.
5. The paper says it adopts VAD as the backbone architecture and retains the same training objectives outside the PGS paradigm. Nevertheless, the ablation study in Section 4.4 shows a performance gap between VAD-Base and PGS_Base, even though they are trained with identical losses. What causes this discrepancy?

**Ethical Concerns:**

["NO or VERY MINOR ethics concerns only"]

**Final Justification:**

I will maintain my original rating. The authors have proposed an approach to improve existing end-to-end autonomous driving methods and have demonstrated its effectiveness through experiments. As some other reviewers noted, questions remain regarding its applicability to real-world scenarios, and there may be debate over whether the method fundamentally resolves the causal confusion issue. Nonetheless, I continue to hold a positive overall view of the work and will therefore keep my initial rating.

**Limitations:**

The authors adequately discuss the limitations in the paper and throughout the work. They point out that their proposed framework is heavily dependent on perception accuracy and generalization, and list this reliance as one of its key limitations.

**Paper Formatting Concerns:**

There are no major formatting issues in the paper.

**Quality:**

3

**Strengths And Weaknesses:**

## Strengths
- The paper is clearly written overall, and the figures and equations are well organized, making the proposed framework easy to understand.
- The experiments demonstrate higher task success rates than previous studies.
- The authors include extensive visualizations of the results and provide a thorough analysis of limitations and failure cases.

## Weaknesses
- While the related works discussed in Section 2.3 and studies such as [1] encourage networks to autonomously discover causal cues directly from data, the proposed PGS framework introduces auxiliary losses based on perception module outputs, such as lane centerlines and the predicted future trajectories of surrounding objects, to inject elements that are considered important from a human perspective, like lane keeping and collision avoidance. Although these elements are not implemented as manually crafted rules or explicit filters, but rather are incorporated into the learning process through network training, and can be seen as a practical compromise to reduce imitation noise and improve safety, I am somewhat concerned that this explicit intervention may diminish the strength of the work in comparison to purely end-to-end imitation learning approaches.
- I believe the paper’s contribution could be emphasized further by adding experiments in environments built on real-world datasets such as nuScenes [2], in addition to the CARLA simulation where surrounding vehicles behave ideally.

References:

- [1] De Haan, Pim, Dinesh Jayaraman, and Sergey Levine. "Causal confusion in imitation learning." *Advances in neural information processing systems* 32 (2019).
- [2] Caesar, Holger, et al. "nuscenes: A multimodal dataset for autonomous driving." *Proceedings of the IEEE/CVF conference on computer vision and pattern recognition*. 2020.

---

> ### Author Rebuttal · Authors · 2025-07-31
>
> ## Response to Reviewer gAQK.
> > **Q1. Do this explicit intervention may diminish the strength of the work in comparison to purely end-to-end imitation learning approaches?**
>
> **A1.** Thank you for your insightful feedback. While our PGS framework introduces auxiliary losses derived from perception outputs—such as lane geometry or predicted trajectories—it remains a fully trainable end‑to‑end model. All perception, planning, and policy modules are jointly optimized without introducing hard-coded or frozen components.
>
> This approach offers a principled way to improve safety and reduce imitation noise by guiding the network to focus on causal features (e.g., lane following and collision avoidance cues), without sacrificing flexibility or learning capacity.
>
> By contrast, purely end-to-end imitation often encounters causal confusion, where models learn spurious correlations rather than underlying causal mechanisms—especially under distributional shift. As empirically documented, more observational inputs can indeed degrade performance if the model incorrectly infers cause from effect. The auxiliary supervision in PGS can help mitigate such confusion by nudging the model toward robust, causally relevant patterns during training.
>
> > **Q2. Adding experiments in environments built on real-world datasets such as nuScenes will be ideally.**
>
> **A2.** Thank you for this valuable recommendation. We fully acknowledge the significance of evaluating autonomous driving methods on real-world datasets. Indeed, nuScenes is a highly influential benchmark with rich annotations that has notably advanced research in perception and prediction. However, for assessing the performance of end-to-end driving systems, nuScenes exhibits several key limitations:
>
> Firstly, nuScenes provides primarily open-loop data. Recent studies (e.g., Reference [19]) have explicitly demonstrated that open-loop performance often fails to reflect realistic closed-loop capabilities accurately. Remarkably, simple baseline models—such as a single-layer MLP conditioned only on the ego vehicle’s current state—can achieve superficially strong open-loop performance metrics on nuScenes, underscoring this limitation.
>
> Secondly, nuScenes predominantly comprises simple, straight-driving scenarios, which constrains its capacity to probe causal reasoning and to evaluate generalization in rare yet safety-critical situations.
>
> In contrast, our work focuses explicitly on improving closed-loop performance in interactive and high-risk settings. Bench2Drive encompasses a wide range of driving scenarios and offers relatively comprehensive evaluation metrics, which constitutes the primary reason for selecting it as our main evaluation benchmark.
>
> Lastly, we sincerely hope to further validate and extend our method on benchmarks that combine real-world data with closed-loop evaluation.As part of our future work, we plan to explore real-world evaluation using Bench2Drive-R, which provides real-world driving data with richer map annotations and supports closed-loop benchmarking. We believe this will offer a more principled and reliable framework for evaluating decision-making performance in the real world.
>
> > **Q3. Could PGS also be leveraged for the branches that output object trajectories and mode-classification scores?**
>
> **A3.** Thank you for your valuable suggestion. The proposed PGS framework can indeed be extended to the mode-classification branch, as it shares similar decision-level semantics with ego planning.
>
> However, applying PGS directly to object trajectory prediction is less straightforward. In ego planning, the future intention (e.g., turning left or right in form of target ponint) is clearly defined and remains applicable during both training and deployment phases. In contrast, for surrounding objects, such intent information is typically unavailable or ambiguous, especially at intersections. To effectively apply PGS in this context, additional intent estimation or behavior modeling would be required. We consider this a promising direction for future work. We believe exploring such extensions could further enhance the model’s holistic scene understanding and planning reliability.
>
> > **Q4. What does $\beta$ represent in line 240?**
>
> **A4.** $\beta$ represents the threshold distance used to determine potential collisions. In our model, we set $\beta$ = 0.5 meters to define a safe margin between agents.
>
> > **Q5. What accounts for the imbalance of high driving-score and success-rate and lower scores for efficiency and comfort?**
>
> **A5.** Thank you for your thoughtful question. Please refer to our response to Reviewer qCVeQ, Q2 for a detailed explanation.
>
> > **Q6. Are “PPGSS” and “PGS” in Tables 1 and 2 the same?**
>
> **A6.** We really appreciate your attention to detail. Yes, "PPGSS" is a typo and will be corrected to "PGS" in the revised version.
>
> > **Q7. Why there is a performance gap between VAD-Base and PGS_Base?**
>
> **A7.** Thank you for your thoughtful question. As stated in line 154 of the main paper, the proposed $ PGS_{base}$ model does not adopt the cascaded prediction–decision architecture used in VAD. Instead, it employs a unified decoder for both ego trajectory planning and object motion prediction. This architectural difference inherently introduces changes in representation and decision coupling.
>
> Furthermore, we would like to offer an important clarification. Our model does not incorporate the auxiliary planning losses used in the VAD framework. The original intent was to adopt the full loss suite from VAD’s perception module only. We sincerely apologize for the confusion and will revise the manuscript to clearly state this distinction.

---

> > ### Comment · Reviewer_gAQK · 2025-08-05
> >
> > Thank you for your response. I feel that most of my questions have been sufficiently addressed. In particular, I appreciate the clarification regarding the evaluation on nuScenes. I had considered raising further discussion related to real-world applications, but since Reviewer RHTg has already brought this up, I will refer to the comments under their review instead.

---

> > > ### Author Response · Authors · 2025-08-05
> > >
> > > Thank you for your kind follow-up and for acknowledging our clarifications. We sincerely appreciate your thoughtful engagement with our work. We have been engaged in further discussion with Reviewer RHTg regarding the issue of real-world transferability, and we hope that those exchanges help to clarify your concerns as well. Should you have additional thoughts or suggestions arising from that discussion, we would be grateful to receive any further questions or feedback.

---

> > > ### Author Response · Authors · 2025-08-08
> > >
> > > Upon further reflection on our discussion regarding your Q1, we would like to provide a more precise clarification, as we feel our initial response may not have fully addressed your underlying concern.
> > >
> > > As you pointed out, there is significant momentum in industry toward enabling end-to-end networks to directly learn causal relationships through data scaling and expanded model capacity. This approach draws strong validation from the success of large language models, which demonstrate that sufficient scale can enable models to acquire generalizable knowledge. Similarly, the concept of autonomous driving world models has been widely regarded as a potential breakthrough that could accelerate development in this domain.
> > >
> > > However, we identify a fundamental distinction between language and autonomous driving domains. Driving datasets lack the explicit causal markers that enable language model success. Text corpora contain abundant linguistic cues such as 'therefore,' 'thus,' 'because,' 'hence,' 'since,' 'consequently,' and 'lead to' that directly signal causal relationships, whereas driving datasets rarely provide equivalent representations of causality. These markers are essential for helping models learn robust causal patterns from noisy inputs.
> > >
> > > Our PGS framework addresses this gap by leveraging perception outputs—lane geometry and predicted trajectories—to inject structured causal knowledge about lane-keeping and collision avoidance through auxiliary supervision. This approach complements rather than contradicts the scaling paradigm. We believe that relying solely on network capacity and large-scale data to implicitly learn causal knowledge is insufficient. Early end-to-end models like NVIDIA's direct image-to-control mapping demonstrated poor generalization despite large datasets, while subsequent cascaded frameworks such as VAD and UniAD achieved superior results by introducing intermediate perception and prediction tasks. This improvement stems from enhanced causal relationship modeling: intermediate tasks help the planning module filter out irrelevant factors from raw sensor inputs, thereby enabling the planning module to learn causal patterns between driving behaviors and relevant environmental factors.
> > >
> > > We contend that in safety-critical domains, the injection of causal knowledge through auxiliary supervision is not only compatible with end-to-end learning principles but essential for success. Rather than weakening generalization capabilities, this design serves as a necessary strategy to mitigate imitation noise and make scaling both feasible and effective.

---

> > > > ### Comment · Reviewer_gAQK · 2025-08-09
> > > >
> > > > Thank you for the detailed follow-up response. While there may be room for debate on whether the proposed idea is truly groundbreaking, I find it to be a solid and well-executed piece of work. I hold a generally positive view of the paper and will therefore maintain my original rating.

---

### Official Review · Reviewer_RHTg · 2025-07-02

**Clarity:** 3
**Significance:** 3
**Originality:** 3
**Rating:** 4
**Confidence:** 4

**Summary:**

This paper introduces PGS, a new training paradigm for end-to-end autonomous driving. Rather than relying solely on expert trajectories for supervision, the authors propose using structured and interpretable perception outputs—such as lane centerlines and predicted motions of other agents—as both inputs and supervisory signals for decision-making modules. Three self-supervision losses are designed. Experimental results on the challenging Bench2Drive benchmark demonstrate that this perception-aligned self-supervised approach outperforms sota methods.

**Questions:**

See weaknesses.

**Ethical Concerns:**

["NO or VERY MINOR ethics concerns only"]

**Final Justification:**

I assume the authors will incorporate this core discussion of real-world transferability into future versions of the manuscript (if the paper is accepted).

Authors  response fully address my concerns, and I  maintain my positive rating

**Limitations:**

Yes.

**Quality:**

3

**Strengths And Weaknesses:**

Strenths:
1. The PGS approach is implemented via three well-defined loss functions (MTPS, STPS, NTPS) that are explained in detail mathematically and graphically.
2. The approach is evaluated on the g closed-loop benchmark, which is more realistic than common open-loop metrics. Table 1 shows significant quantitative improvements in driving scores and success rates. Table 3 eliminates the effects of each component. The authors provide more examples in the supplementary material.
3. The core idea of ​​PGS does not depend on certain architectures; instead, it is compatible with standard BEV-based processes and can be generalized to other frameworks.

Weaknesses:
- There seems to be no empirical evaluation of robustness to noise or imperfect perception, which may occur in real applications. For example, in safety-critical scenarios, does the system performance collapse if the perception system fails to detect some vehicles?

- Did the authors try to implement and test PGS in any real-world datasets/simulations (there are indeed some recent works that perform closed-loop simulations on real-world scenarios, i.e. DriveArena, NavSim)?

---

> ### Author Rebuttal · Authors · 2025-07-31
>
> ## Response to Reviewer RHTg.
>
> > **Q1. Does the system performance collapse if the perception system fails to detect some vehicles?**
>
> **A1.** Thank you for your valuable question. This is indeed a critical consideration, especially in the context of real-world deployment. A similar concern was also raised by Reviewer M41J, Q1, to which we have provided a detailed response. We kindly refer you to that reply for a comprehensive discussion of this issue.
>
> > **Q2. Did the authors try to implement and test PGS in any real-world datasets/simulations such as DriveArena, NavSim?**
>
> **A2.** Thank you for your valuable suggestion. We have carefully considered your comment. DriveArena is currently primarily used for data generation and reinforcement learning in closed-loop training settings. In contrast, our method is specifically developed for training on offline datasets—collected from real vehicles—to enhance closed-loop performance via novel algorithmic and architectural improvements. Indeed, training on off-line datasets remains the predominant paradigm for deploying end-to-end autonomous driving models both in academia and industry, particularly due to the practical constraints associated with online training. Furthermore, DriveArena currently lacks standardized evaluation protocols and publicly available benchmarks, which makes it difficult to adopt at this stage. Nonetheless, we fully agree that closed-loop training represents a promising future research direction, and we intend to explore this in subsequent extensions of our work. Regarding experiments on the NavSim, please refer to our response to Reviewer M41J, Q5.

---

> > ### Comment · Reviewer_RHTg · 2025-08-04
> >
> > Thank you for your response.
> >
> > I acknowledge your work showing strong performance on the CARLA-based Bench2Drive benchmark, along with the points regarding the current limitations of DriveArena and NavSim for your research scope.
> >
> > I note that both I and Reviewer M41J raised the question regarding testing PGS on real-world datasets or in more realistic simulation environments.  Given that the ultimate goal of autonomous driving is real-world deployment, could you please provide some insight into the PGS paradigm, validated in the CARLA simulation engine, can be expected to transfer effectively to the real world?

---

> > > ### Author Response · Authors · 2025-08-05
> > >
> > > Thank you for carefully reading our discussion on the limitations of current real-world datasets in validating our method, and for raising a further, more specific question.
> > >
> > > First, we wish to reiterate that the core contribution of the PGS framework lies in its novel use of perception outputs as auxiliary supervision signals to strengthen causal reasoning within the planning module. Technically speaking, the perception capabilities underpinning PGS—such as lane centerline detection and multi-agent trajectory prediction—have already been robustly validated in several leading real-world benchmarks. For instance, HDMapNet and CenterLineDet have demonstrated state-of-the-art lane-centerline detection performance on the nuScenes dataset. Similarly, established trajectory prediction methods like HiVT, MultiPath++, and Scene Transformer have achieved top-ranking results across prominent real-world benchmarks, including nuScenes, Waymo Open, and Argoverse. Furthermore, we emphasize that the proposed PGS framework is entirely simulation-agnostic during both training and deployment, employing no CARLA-specific assumptions or constraints. Thus, from a technical perspective, no fundamental barriers exist that would impede extending the PGS approach to real-world datasets.
> > >
> > > Second, we would like to contextualize this transferability concern within the broader landscape of recent advances in end-to-end autonomous driving research. Two important trends emerge:
> > >
> > > (a) Methods initially validated through open-loop evaluation on real-world datasets often exhibit notable performance degradation when tested in closed-loop scenarios. For example, both VAD and UniAD have reported state-of-the-art open-loop prediction results on the nuScenes benchmark, yet their performances significantly deteriorate when directly evaluated in closed-loop simulation environments such as Bench2Drive. Addressing this discrepancy, subsequent variants (e.g., VADv2) required substantial modifications in network architecture and planning strategies to recover performance under closed-loop conditions (CARLA Town05 benchmark), underscoring inherent limitations in purely open-loop validation frameworks.
> > >
> > > (b) Conversely, models originally developed under closed-loop simulation environments have demonstrated strong transferability to real-world datasets. A representative example is TransFuser (CVPR 2021), a compact, closed-loop simulation-based model that initially achieved top performance on the CARLA Leaderboard. TransFuser was later successfully transferred to real-world settings (e.g., the NAVSIM dataset), surpassing substantially larger-scale models such as UniAD and ParaDrive in both efficiency and effectiveness. In fact, over the past three years, most end-to-end models evaluated on nuScenes, Waymo, or NAVSIM have adopted TransFuser as a lightweight baseline, and its architecture has been widely extended in subsequent real-world systems. For instance, the current state-of-the-art NAVSIM model, GoalFlow, inherits its perception module from TransFuser. Moreover, the recent WoTE framework (ICCV 2025), like PGS, employed the Bench2Drive benchmark for closed-loop validation and was also evaluated on the real-world NAVSIM dataset, achieving state-of-the-art performance on both. This highlights that rigorous closed-loop evaluation in realistic simulation environments can effectively support strong generalization to real-world scenarios.
> > >
> > > In summary, both from a technical standpoint and based on the transfer patterns observed in prior work, we believe that models developed with simulated data and validated under closed-loop settings generally retain their effectiveness when deployed in real-world scenarios. Accordingly, we are confident that PGS can achieve similarly robust results in real-world settings.

---

> > > > ### Comment · Reviewer_RHTg · 2025-08-07
> > > >
> > > > Thank you for the detailed explanation and further clarification. Your discussion on the limitations of open-loop versus closed-loop validation is particularly insightful and resonates with me.
> > > >
> > > > I assume the authors will incorporate this core discussion of real-world transferability into future versions of the manuscript (if the paper is accepted).
> > > >
> > > > Your response fully addresses my concerns, and I have no further questions.

---

> > > > > ### Author Response · Authors · 2025-08-07
> > > > >
> > > > > Thank you for highlighting this important aspect. We will incorporate this valuable discussion on real-world transferability into the supplementary material.  We sincerely appreciate you raising this question!

---

### Official Review · Reviewer_M41J · 2025-07-03

**Clarity:** 3
**Significance:** 2
**Originality:** 3
**Rating:** 4
**Confidence:** 4

**Summary:**

This paper proposes a new training paradigm for end-to-end autonomous driving, termed Perception-Guided Self-Supervision (PGS), which aims to mitigate causal confusion—a major limitation of imitation learning—by replacing expert trajectories with structured outputs from perception modules as supervision signals. The authors introduce three types of supervision:
MTPS: Multi-Modal Trajectory Planning Self-Supervision, which reframes trajectory prediction as a lane selection problem.
STPS: Spatial Trajectory Planning Self-Supervision, which aligns the predicted trajectory with lane centerlines to avoid noisy human demonstration.
NTPS: Negative Trajectory Planning Self-Supervision, which penalizes ego trajectories that intersect with future predicted trajectories of surrounding agents.
Extensive experiments on the Bench2Drive closed-loop benchmark demonstrate that the proposed method outperforms prior state-of-the-art approaches, achieving significant improvements in both overall driving score and critical scenario success rate.

**Questions:**

1. Will the code be released? Open-sourcing the implementation would help the community reproduce and build upon the proposed method.

2. Since the method relies heavily on perception outputs as supervision, how does it perform when perception results are inaccurate or noisy?

3. Have the authors considered evaluating the method on real-world datasets such as NAVSIM to validate its applicability beyond simulation?

4. Typo in Figure 1: Should "Limitation learning" be "Imitation learning"? Please clarify and correct if necessary.

**Ethical Concerns:**

["NO or VERY MINOR ethics concerns only"]

**Final Justification:**

My concerns are mostly resolved, so I will maintain a positive rating. I agree with Reviewer RHTg and look forward to seeing results on a real-world dataset, such as NAVSIM, in the revised manuscript.

**Limitations:**

yes

**Quality:**

3

**Strengths And Weaknesses:**

Strengths
1. The paper addresses the key challenge of causal confusion in end-to-end driving—a well-known problem that limits real-world deployment. The motivation is clear and well justified.

2. The use of structured perception outputs (e.g., lane centerlines, predicted agent motion) as training supervision is intuitive, principled, and avoids the pitfalls of noisy expert demonstrations.

Weaknesses:
1. The proposed paradigm heavily depends on accurate perception outputs (e.g., lane centerlines, object trajectories). While this is acknowledged in the limitations, it raises concerns about robustness under real-world perception errors, especially in long-tail or rare scenarios.

2. Terminology of “self-supervision” may be misleading. The term “self-supervision” typically refers to learning from raw data without any external labels. In this work, supervision is provided by another part of the network (perception), which is trained with labeled data. Some readers may find this use of the term confusing or inconsistent with common definitions.

---

> ### Author Rebuttal · Authors · 2025-07-31
>
> ## Response to Reviewer M41J.
>
> > **Q1. What happens if the perception module fails catastrophically?**
>
> **A1.** We appreciate your thoughtful comment. We fully recognize that our framework depends on perception outputs, which naturally raises questions about robustness under perception errors. We would like to clarify our motivation, underlying assumptions, and the scope of our contributions in relation to this concern.
>
> The design of our paradigm is motivated by a key empirical insight: in existing end-to-end driving systems, perception is rarely the dominant performance bottleneck. In the majority of scenarios, modern perception modules already provide sufficiently accurate outputs for downstream tasks. Despite this, end-to-end systems often suffer from causal confusion in their decision-making process—where models latch onto spurious correlations rather than semantically meaningful or causally relevant signals.
>
> Based on this observation, the primary objective of our work is to improve causal modeling within the planning and control module and to verify its impact on the performance of end-to-end systems. And we believe that the proposed improvements can enhance ego vehicle behavior in most scenarios where perception is effective.
>
> We acknowledge that, like all perception-dependent systems, our method inherits the limitations of upstream perception. In rare or long-tail scenarios where perception fails, performance degradation is expected. However, this sensitivity is not unique to our method; rather, it is an intrinsic property of any perception-dependent autonomous driving approach—whether end-to-end or modular. Errors originating from the perception module naturally propagate downstream, cumulatively affecting planning and control decisions.
>
> Crucially, our framework neither introduces additional vulnerabilities nor exacerbates existing issues beyond those inherently present in conventional perception-driven approaches. In fact, enhancements in perception robustness directly translate into improved performance for our approach. Therefore, we see our contribution as complementary and orthogonal to existing perception improvement efforts.
>
> In summary, while acknowledging the inherent sensitivity to perception quality, we emphasize that our method primarily targets the critical issue of causal confusion—demonstrably improving system robustness under standard perception conditions. Improving perception robustness itself remains a broader and longstanding goal in the autonomous driving research community, and advancements in this area would directly benefit the effectiveness of our proposed framework.
>
> > **Q2. The usage of the term “self-supervision” may be inconsistent with its conventional definition, as the supervision in this work comes from perception outputs trained with labels.**
>
> **A2.** Thank you for raising this insightful question. We had indeed discussed this point extensively prior to the submission, as your description precisely captures. We ultimately chose to use the term self-supervision because the main contributions of our work are organized around a novel learning paradigm for the planning and control module. This terminology was intended to emphasize the conceptual distinction from existing imitation learning paradigms. As a result, we finalized the term perception guided self-supervision after careful deliberation.
>
> This term is meant to clarify the differences between our approach and standard definitions of self-supervision: specifically, our supervisory signals are derived from an on-the-shelf perception module. We will revise the manuscript to clarify the use of this terminology and more explicitly articulate its intended meaning, in order to avoid potential confusion.
>
> > **Q3. Will the code be released?**
>
> **A3.** Yes, we fully agree that open-sourcing the implementation would benefit the community by facilitating reproducibility and further research. We will release the code upon acceptance of the paper.
>
> >**Q4. How does the method perform when perception results are inaccurate or noisy?**
>
> Thank you for your question. This concern closely aligns with the point raised in Weakness 1 of your comment. For a detailed response, please refer to our answer to Q1, where we provide a comprehensive discussion addressing this issue.
>
> > **Q5. Have the authors considered evaluating the method on real-world datasets such as NAVSIM?**
>
> **A5.** Thank you for your valuable suggestion.First, existing studies have demonstrated that although methods evaluated under open-loop settings may yield promising L2 error and collision rates, such models often exhibit significant performance degradation when deployed in real-world scenarios or evaluated under closed-loop settings. Therefore, our work focuses primarily on enhancing end-to-end model performance under closed-loop evaluation, which we believe holds greater practical value for autonomous driving systems.
>
> For this reason, we chose Bench2Drive (B2D) as our primary benchmark. On one hand, this dataset encompasses a wide range of complex traffic scenarios, including 44 interactive categories such as _ParkingExit, ParkingCutIn, ParkedObstacleTwoWays, and HazardAtSideLane_, along with 23 different weather conditions (e.g., _sunny, foggy, rainy_), effectively covering the majority of real-world driving contexts. The evaluation set ensures comprehensive coverage across all categories. On the other hand, the evaluation is conducted in a fully closed-loop manner—meaning the behavior of surrounding agents dynamically responds to the ego vehicle’s actions—thereby better aligning with the dynamics of real-world deployment.
>
> In contrast, the NAVSIM dataset is derived from nuPlan, which provides approximately 1,200 hours of human driving data. However, it does not directly include adverse conditions such as heavy rain or nighttime driving.  Furthermore, its evaluation is semi-closed-loop—or arguably still open-loop—as the behavior of other agents does not react to the ego vehicle's decisions. This makes the evaluation metrics insufficient to fully reflect the algorithm’s performance in real-world closed-loop conditions.
>
> Second, the core motivation of our work is to address the causal confusion issues that frequently arise in closed-loop or real-world testing. A classic example of such confusion is observed in how autonomous vehicles behave at traffic light intersections: models suffering from causal confusion fail to learn the correct association between traffic light signals (red/green) and the ego vehicle's actions, instead basing their decisions on the behavior of surrounding vehicles. This represents one of the most critical and realistic forms of causal confusion, which is not captured in NAVSIM's evaluation metrics. NAVSIM focuses solely on metrics such as collision rate, drivable area violations, efficiency, and target goal achievement, without assessing whether the model correctly understands the causal relationship between traffic signals and ego behavior at traffic lights or stop lines. In contrast, the B2D benchmark explicitly reflects this type of causal failure: confusion regarding traffic light cues can lead to vehicles becoming stuck or violating traffic regulations, ultimately resulting in task failure. Furthermore, B2D includes a dedicated set of scenarios involving interactions with traffic signals and stop signs (i.e., the “traffic sign” scenario group), which allows for a more targeted and comprehensive evaluation of a model’s causal reasoning capabilities.
>
> Thus, although B2D may have a larger domain gap from real-world data in terms of visual appearance and sensor modality, we argue that from an algorithmic perspective, B2D and its evaluation protocol more accurately reflect real-world performance.
>
> In summary, due to the comprehensiveness of B2D's scenario coverage, its alignment with real-world deployment dynamics, and its ability to reflect the most challenging causal reasoning problems in end-to-end driving, we selected B2D as a more suitable and challenging benchmark. In contrast, NAVSIM may not be well-suited for addressing the specific issues our work targets.
>
> Lastly, we sincerely hope to further validate and extend our method on benchmarks that combine real-world data with closed-loop evaluation. We have assessed the feasibility of adapting our method to NAVSIM, and found it would involve substantial effort, including dataset format conversion, constructing intermediate supervision signals (e.g., verifying whether NAVSIM includes lane-level annotations such as lane centerlines), validating perception-guided self-supervised signals (e.g., tuning thresholds for relevant lane filtering modules, and visual verification of MTPS, STPS, and NTPS losses). Given the limited time available during the rebuttal phase, it is technically infeasible to robustly adapt our method to NAVSIM within this timeframe.
>
> Luckily, we have noted that the B2D team announced in December 2024 the upcoming release of an extended version—Bench2Drive-R—which will include both real-world and simulation data. This extended benchmark will better fulfill both data-level and evaluation-level alignment with real-world conditions. We plan to validate our method on this dataset as soon as it becomes publicly available.
>
> > **Q6. Should "Limitation learning" be "Imitation learning" in Figure 1?**
>
> **A6.** Thank you for your careful review. Yes, this is a written error and should be corrected to “Imitation learning.” We will fix it in the revised version.

---

> > ### Comment · Reviewer_M41J · 2025-08-07
> >
> > Thank you for your response. I agree with Reviewer RHTg and look forward to seeing results on a real-world dataset, such as NAVSIM, in the revised manuscript.

---

> > > ### Author Response · Authors · 2025-08-07
> > >
> > > Thank you for your continued engagement and for emphasizing the importance of evaluating real-world transferability. We agree that this is a key aspect for end-to-end autonomous driving systems.
> > >
> > > We would like to clarify that we have already addressed two key aspects of this concern in our previous responses:
> > >
> > > First, regarding the use of NAVSIM, we respectfully noted that its scenario distribution and evaluation protocol are not well aligned with the causal confusion issues in ego planning that PGS is specifically designed to address. In contrast, Bench2Drive offers richer interactive scenarios and a closed-loop setup that better reflect the causal challenges targeted by PGS. A detailed explanation is provided in our response to your Q5.
> > >
> > > Second, Reviewer RHTg raised the broader question of whether models trained in simulation can effectively transfer to real-world settings. We articulated our perspective on the transferability of PGS based on both technical considerations and recent trends in the performance of autonomous driving approaches. Reviewer RHTg acknowledged and agreed with this perspective. For further context, we kindly refer you to our discussion with Reviewer RHTg.
> > >
> > > Finally, we hope these clarifications help address your concerns regarding real-world applicability. Should you have any additional questions or suggestions, we would be more than happy to continue the discussion.

---

### Official Review · Reviewer_qCVe · 2025-07-07

**Clarity:** 2
**Significance:** 3
**Originality:** 3
**Rating:** 4
**Confidence:** 2

**Summary:**

The paper presents an algorithm to improve the popular imitation strategy for autonomous driving.  Specifically, it is argued that the conventional imitation learning strategy for autonomous driving is agnostic to the errors of safety concerns when minimizing the L2 loss during behavior cloning. This leads to critical safety errors in the output results. To address this, the paper introduces three kinds of losses during training to suppress such errors. Two of them aims to provide better target lane binding by quantizing the expert trajectory into lane-specific templates, and the third penalises the ego vehicle trajectory that collides with other dynamic road objects. Experiments on a closed-loop benchmark show that the proposed method can outperform other baselines in most scenarios under closed-loop metrics.

**Questions:**

- Will the quantisation of STPS cause some discontinuity in the target trajectory? How does that affect the performance?

**Ethical Concerns:**

["NO or VERY MINOR ethics concerns only"]

**Final Justification:**

The rebuttal addresses most my initial concerns and questions. I keep my original score.

**Limitations:**

See Strengths And Weaknesses.

**Quality:**

3

**Strengths And Weaknesses:**

-  Strengths
   - Interesting observation and motivation on the safety errors of behavior cloning for autonomous driving.
   - Reasonable technical solutions and execution.
   - Results look promising, with improvements over the baselines and other state-of-the-art methods with clear margins.

- Weaknesses
  - Some interesting cases are not tested. For instance, the first two introduced losses relies on map elements. What about unstructured cases? If would be helpful to show how PGS with only NTPS works in such cases.
  - Some performance metric drops are not explained. It seems that the introduced losses may cause the efficiency and comforness degrade significantly in Table 1. What are the reasons for that? Shouldn't the inference cost of the proposed method the same as the VAD-base?
  - Presentation, some notations are not clear or defined yet. E.g., what is the plus sign in $Q_{motion}$ between Lien 159 and Line 160? "Limitation learning" in Figure 1? It would also be helpful to provide some illustration for the quantities defined for section 3.1 through 3.4. Crucial equations should be numbered for easy reference too.

---

> ### Author Rebuttal · Authors · 2025-07-31
>
> ## Response to Reviewer qCVe.
>
> > **Q1. For unstructured cases, show how PGS with only NTPS works.**
>
> **A1.** This is a reasonable suggestion, and thank you for pointing out our oversight. We have conducted additional experiments specifically addressing your concern by training a variant of our model (denoted as $PGS_{NTPS}$) using only the NTPS loss, thus entirely removing dependencies on structured map elements. As summarized in Table 1, compared to the complete PGS model, the NTPS-only variant demonstrates significantly improved performance in merging scenarios. This result can be attributed to NTPS’s strong guidance in collision-prone, complex intersection scenarios. Conversely, performance slightly decreased in overtaking scenarios, as NTPS may lead to overly conservative behavior when static obstacles are present in the ego lane. Upon further incorporating MTPS and STPS losses, structured road priors provide richer spatial guidance, effectively enlarging the decision space. This allows the model to learn more flexible maneuver patterns, such as overtaking by borrowing adjacent or opposing lanes. Consequently, the overall success rates improve in both _merging_ and _overtaking_ scenarios. Both quantitative and qualitative analyses confirm our original hypothesis: incorporating structured road priors significantly strengthens causal reasoning, enhancing decision robustness under structured driving conditions. In unstructured environments where road priors are unavailable, we expect the full PGS model to behave similarly to the NTPS-only variant.
>
> **Table 1: Ablation Study of the Proposed PGS Framework with NTPS-only PGS Model.**
>
> | **Method**             | **Avg. L2** | **Merging (%)** | **Overtaking (%)** | **Mean (%)** |
> | ---------------------- | ----------- | --------------- | ------------------ | ------------ |
> | VAD-Base               | 0.91        | 8.11            | 24.44              | 16.28        |
> | VAD-Tiny               | 1.15        | 9.33            | 11.11              | 10.22        |
> | $PGS_{Base}$           | 0.87        | 16.46           | 13.33              | 14.89        |
> | **$PGS_{NTPS}$**       | **0.90**    | **25.00**       | **6.67**           | **15.84**    |
> | $PGS_{Base+STPS}$      | 0.78        | 24.44           | 26.25              | 25.35        |
> | $PGS_{Base+STPS+MTPS}$ | 0.75        | 23.75           | 44.44              | 34.10        |
> | $PGS_{All}$            | 0.77        | 32.50           | 66.67              | 49.58        |
> | $PGS_{self}$           | 2.89        | 31.25           | 35.56              | 33.40        |
> > **Q2. Why the introduced losses may cause the efficiency and comforness degrade significantly in Table 1? Shouldn't the inference cost of the proposed method the same as the VAD-base?**
>
> **A2.** Thank you very much for this thoughtful comment. To clarify, the _efficiency_ metric is computed based on comparisons between the ego vehicle’s speed and that of surrounding traffic, rather than the inference cost. This terminology might have caused confusion. Since our method shares similar architectural complexity with VAD, it does not introduce additional computational overhead during inference. The _comfortness_ metric, on the other hand, is derived from acceleration and jerk measurements.
>
> First of all, prompted by your constructive question, we identified a critical issue in our evaluation process. Specifically, we inadvertently used an outdated version of the official `atomic_criteria.py` file (located at `Bench2Drive/scenario_runner/srunner/scenariomanager/scenarioatomics/atomic_criteria.py`). This earlier version incorrectly computed the efficiency metric using only cases with minimum-speed infractions (i.e., speed < 100%), while entirely omitting scenarios labeled as "Perfect" (i.e., completed without speed violations). This issue is confirmed by the JSON files included in our supplementary materials (`b.PGS_all_metric_files`).
>
> The latest version of the official script, which we have now adopted, correctly follows the evaluation protocol described in the original paper by recording speed measurements at every 5% interval along the route to ensure comprehensive and fair metric computation.
>
> Under the corrected protocol, our updated Efficiency and Comfortness scores are 180.9974 and 12.0039, respectively. Notably, our model achieves a new **state-of-the-art** result in Efficiency. These values will be updated in the final version of the paper.
>
> However, we would like to emphasize that Efficiency and Comfortness should be interpreted under comparable scenario success rates to ensure fair and meaningful comparisons. For example, if a vehicle becomes stuck at an intersection and fails to complete the scenario, the computed Comfortness may be deceptively high due to the lack of motion. Conversely, a model that completes the scenario at high speed—but with a collision—may report high Efficiency despite safety violations. Therefore, we argue that Driving Score and Success Rate are more indicative of real-world performance, and comparisons of secondary metrics such as Efficiency or Comfortness should be contextualized under similar success conditions.
>
> Secondly, in challenging driving scenarios such as overtaking by temporarily borrowing the opposing lane, our model prioritizes safety, often resulting in conservative maneuvers. Specifically, the model tends to decelerate or come to a full stop before accelerating rapidly to complete the maneuver safely. While this behavior closely resembles cautious human driving, it inevitably leads to increased acceleration and jerk, thereby negatively impacting the comfort score.
>
> Moreover, our method leverages lane-centerline-based pseudo trajectories as supervision signals. These pseudo trajectories effectively guide the model to adhere more accurately to road topology at intersections, thereby reducing erroneous lane choices. However, because these pseudo labels prioritize spatial correctness over temporal continuity, the resulting trajectories may exhibit slight discontinuities compared to human expert demonstrations, which further degrades the comfort metric. Addressing this temporal discontinuity to improve trajectory smoothness is a key part of our future work.
>
>
> > **Q3. Issues with notation clarity and paper formatting.**
>
> **A3.** We appreciate your careful review. We will revise the manuscript accordingly to improve the clarity of formatting, as per your suggestion. Specifically, the "⊕" sign indicates concatenation between features, The term “Limitation learning” in Figure 1 is a written error and will be corrected to “Imitation learning.”
>
>
> > **Q4. Will the quantisation of STPS cause some discontinuity in the target trajectory? How does that affect the performance?**
>
> **A4.** Your observation is correct. Quantizing spatial trajectories in STPS inevitably introduces minor discontinuities compared to smooth human expert trajectories. These discontinuities manifest as subtle velocity fluctuations, which contribute to reduced comfort. Addressing this concern, we plan to implement spatial smoothing techniques in future iterations to alleviate trajectory discontinuities and thus enhance comfort metrics.

---

> > ### Comment · Reviewer_qCVe · 2025-08-05
> >
> > Thanks for the response to the concerns and questions raised in the original review.
> >
> > >A1: "... In unstructured environments where road priors are unavailable, we expect the full PGS model to behave similarly to the NTPS-only variant."
> >
> > It wound be great if the quantitative results can be included in this case, as it highly reflects the practical deployment scenarios.

---

> > > ### Author Response · Authors · 2025-08-05
> > >
> > > Thank you for your valuable comment and for highlighting the importance of evaluating our method in unstructured environments, which indeed reflect many practical deployment scenarios. We agree that including quantitative results in such settings would provide valuable insights into the generalizability of our approach, and we plan to incorporate these evaluations in a revised version of the paper.

---

### Note · Authors · 2025-08-13

Dear Area Chair,

The reviewers’ insightful comments and constructive discussions have further clarified the main contributions of our paper, most concerns have been meaningfully addressed. Corresponding revisions will be included into the revised manuscript and supplementary materials, and we sincerely appreciate their time and effort in reviewing this work.

Three reviewers questioned the necessity of evaluating on real-world datasets. We clarified that existing benchmarks have fundamental limitations for assessing causal confusion mitigation. nuScenes offers only open-loop evaluation, where models predict trajectories without influencing subsequent inputs, while NAVSIM uses semi-closed-loop evaluation where surrounding agents do not react to ego decisions. Both fail to capture interactive dynamics critical for evaluating causal confusion and lack comprehensive coverage of traffic light and complex intersection scenarios where such issues occur.

This limitation motivated our selection of Bench2Drive over real-world alternatives. We established PGS’s real-world adaptability from two perspectives. Technically, PGS employs no simulation-specific assumptions and relies on perception capabilities validated on real-world benchmarks such as nuScenes, Waymo Open, and Argoverse. Empirically, prior work shows that models validated in rigorous closed-loop simulation—such as CARLA—tend to transfer effectively to real-world deployment, as evidenced by TransFuser and WoTE. In contrast, models validated only on real but open-loop datasets often require extra effort to adapt to closed-loop or real-world settings, as in VADv2.

Our discussions clarified that Bench2Drive provides the most appropriate evaluation framework for PGS’s design objectives while offering sufficient scenario complexity to validate the generalization ability of our core contributions. We have shown that PGS has strong potential to address causal confusion—a longstanding challenge—and are confident in its applicability to real-world scenarios. Future validation on Bench2Drive-R will provide additional empirical support. Most reviewers felt our responses clarified their concerns and aligned with our perspective.

We appreciate your role in evaluating both the technical rigor and the broader impact of this work, which are essential for determining its contribution to the field. We trust your judgment in assessing how PGS advances the development of robust and deployable autonomous driving systems.

---

### Decision · Program_Chairs · 2025-09-17

**Decision:**

Accept (poster)

**Comment:**

This paper introduces Perception-Guided Self-Supervision (PGS), a novel approach to mitigate causal confusion in end-to-end autonomous driving by leveraging perception outputs as supervisory signals. The authors effectively demonstrate the method's potential, achieving state-of-the-art results on the Bench2Drive benchmark and providing strong empirical evidence for its effectiveness. The work addresses a fundamental limitation in imitation learning and offers a principled way to improve causal modeling.

Reviewers highlighted the significance of the problem addressed, the novelty of the approach, and the quantitative improvements achieved.

However, some concerns were raised, primarily concerning the robustness of the method to perception errors in real-world scenarios, the clarity of the term "self-supervision" as used in the paper, and the limited evaluation on real-world datasets. The authors provided detailed responses to these concerns, clarifying their methodology, addressing evaluation issues, and explaining their rationale for benchmark selection. Their explanations regarding real-world transferability and the necessity of closed-loop evaluation for addressing causal confusion were well-received.

Overall, the paper presents a technically sound and significant contribution to the field of autonomous driving. The authors have addressed reviewer concerns, and the proposed PGS framework offers a promising direction for future research. The reviewers' final assessments reflect a positive view of the work.